# Output variability across animals and levels in a motor system

Angela Wenning[1]*, Brian J Norris[1,2], Cengiz Günay[1,3], Daniel Kueh[1], Ronald L Calabrese[1]*

[1]Biology Department, Emory University, Atlanta, United States; [2]Biological Sciences, California State University, San Marcos, United States; [3]School of Science and Technology, Georgia Gwinnett College, Lawrenceville, United States

**Abstract** Rhythmic behaviors vary across individuals. We investigated the sources of this output variability across a motor system, from the central pattern generator (CPG) to the motor plant. In the bilaterally symmetric leech heartbeat system, the CPG orchestrates two coordinations in the bilateral hearts with different intersegmental phase relations ($\Delta\phi$) and periodic side-to-side switches. Population variability is large. We show that the system is precise within a coordination, that differences in repetitions of a coordination contribute little to population output variability, but that differences between bilaterally homologous cells may contribute to some of this variability. Nevertheless, much output variability is likely associated with genetic and life history differences among individuals. Variability of $\Delta\phi$ were coordination-specific: similar at all levels in one, but significantly lower for the motor pattern than the CPG pattern in the other. Mechanisms that transform CPG output to motor neurons may limit output variability in the motor pattern.
DOI: https://doi.org/10.7554/eLife.31123.001

## Introduction

Variability across individuals and across cell types in underlying intrinsic and synaptic properties is now viewed as a hallmark of neuronal networks, even those that produce stereotyped output. Indeed, the hunt is on to find the mechanisms and rules that permit constant network output through coordinated regulation, both developmentally and homeostatically, of highly variable membrane and synaptic conductances (*Davis, 2013*; *Marder et al., 2015*; *O'Leary et al., 2013*, *2014*; *Ransdell et al., 2013*; *Schulz et al., 2006*). But how constant is network output across individuals? Not very seems to be the answer when looking at the literature more closely (e.g., locomotion in mice [*Bellardita and Kiehn, 2015*] and zebrafish [*Masino and Fetcho, 2005*; *Wiggin et al., 2014*]; food processing in crabs [*Hamood et al., 2015*; *Hamood and Marder, 2015*; *Yarger and Stein, 2015*], crawling in fly larvae [*Pulver et al., 2015*]).

The central pattern generating networks of invertebrates have provided some of the best evidence supporting the notion of constant output with underlying variability of conductances (*Goaillard et al., 2009*; *Marder et al., 2015*; *Prinz et al., 2004*; *Ransdell et al., 2013*). Here phase of firing of component neurons is considered a critical aspect of a functional motor pattern, and phase is by no means constant. Even though it is not correlated with period, phase varies considerably across animals as shown in the stomatogastric nervous system (STNS) and in our work on the leech heartbeat system (*Bucher et al., 2005*; *Norris et al., 2006*; *Norris et al., 2007b*; *Wenning et al., 2004a*, *2004b*). Indeed, we were forced to the conclusion that each animal arrives at a unique solution to producing a functional heartbeat motor pattern; based on phase differences in the premotor pattern and synaptic strength patterns from the central pattern generator (CPG) to motor neurons (*Norris et al., 2011*; *Wright and Calabrese, 2011b*). Thus, not only are underlying conductances variable but activity itself is variable and any attempt to elucidate mechanisms of

*For correspondence:
awennin@emory.edu (AW);
ronald.calabrese@emory.edu
(RLC)

Competing interest: See
page 24

Reviewing editor: Jan-Marino
Ramirez, Seattle Children's
Research Institute and University
of Washington, United States

**eLife digest** Many of our everyday behaviors are rhythmic actions, such as walking, breathing and chewing. Networks of neurons called Central Pattern Generators, or CPGs, are in charge of rhythmic behaviors. CPGs send instructions to cells called motor neurons, which in turn tell muscles to contract in a particular sequence to produce rhythmic behaviors.

Rhythmic behaviors follow stereotyped patterns: we recognize walking when we see it. But they also vary between individuals: we can recognize the specific gait or 'walk' of a friend. Wenning et al. set out to discover where this variability in rhythmic behaviors comes from, using the leech heartbeat system as a model. Leeches have two hearts, or more precisely two heart tubes that run along the entire length of the body, one on either side. The two heart tubes beat with different patterns, but under the direction of the CPGs and motor neurons, they swap patterns with each other every few minutes. The CPG neurons that generate these rhythms, the motor neurons that respond, and the heart muscles themselves, i.e. each level of the system, can all be tracked in leeches.

Wenning et al. showed that within each leech, the activity of the CPG neurons, motor neurons and muscles associated with a heart tube varies little. Even when the activity of one of these levels varies less than another, for example between CPG and motor neurons, it is not necessarily reflected in the next level of the system. In some cases, however, variability is seen between opposite sides. Moreover, the rhythmic activity of CPG neurons, motor neurons, and muscle cells in one leech differs greatly from that of another. This likely reflects differences in the genes and life history of the animals.

Wenning et al. provide a roadmap for others to use in identifying sources of variability in rhythmic movements. Applying this approach to existing data sets could help tease apart variability in diverse rhythmic behaviors in a variety of animals.

DOI: https://doi.org/10.7554/eLife.31123.002

regulation must consider what the target limits for regulation are ('What is good enough?'; *Marder et al., 2006*), what are the sources of variability in network output, and how variability at one level in a network influences the variability on another. Here we focus mainly on the latter two questions, having previously established the range of functional output (*Norris et al., 2006*; *Norris et al., 2007a*; *Norris et al., 2007b*; *Wenning et al., 2004a*; *Wenning et al., 2014*).

Leech heartbeat presents an amenable system for answering these questions because all relevant neurons of the CPG and motor neurons are identified and easily recorded, and the motor plant (here the hearts) also can be directly monitored (recent review: *Calabrese et al., 2016*). Moreover, the system is strictly feedforward – CPG to motor neurons, to heart muscle –, its elements are bilaterally symmetrical, it operates without phasic sensory feedback (*Calabrese, 1977*; *Calabrese, 1979*), and it has already been demonstrated to be highly variable in output across individuals at each level. A unique aspect to this system is that the coordination of the CPG, motor neurons, and hearts differs at any given time on the two sides – rear-to-front peristaltic versus synchronous – with periodic switches in coordination between sides (*Norris et al., 2006*; *Norris et al., 2007b*; *Wenning et al., 2004a*; *Wenning et al., 2014*; *Figure 1*, *Figure 1—figure supplement 1* and *Figure 1—video 1*). Thus, while rhythmic circuit output is continuous, it presents episodic coordination states on each body side.

The sources of variability that we considered were (1) inherent variability owing to the stochastic nature of biological processes (analyzing the cycle-to-cycle variabilities within a coordination state episode), (2) repetition variability as the same function is performed multiple times by the same elements (comparing across coordination state episodes), (3) variability within an individual due to differences between genetically identical bilaterally homologous neurons and muscles, and (4) population variability including, but not limited to, genetic variability and variability in individual experience (comparing across animals). We then compare these sources across levels and coordination states.

We show that cycle-to-cycle variabilities in phase were low at all levels, in both coordination states, and on both sides. Thus, activity within an individual is precise. We confirm and quantify the

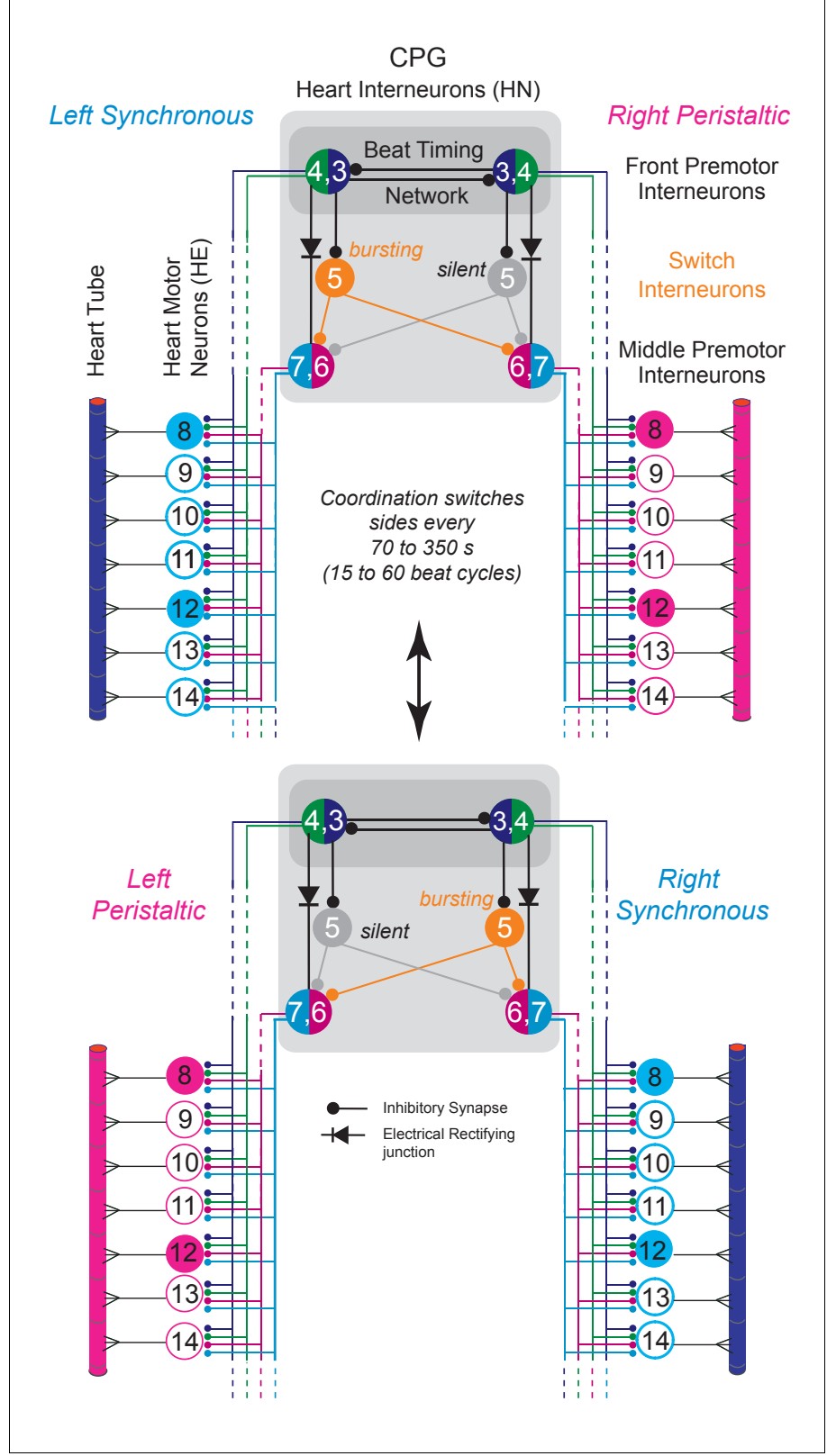

**Figure 1.** The Leech Heartbeat System switches between two coordination states. Circuit diagram including the bilateral homologous pairs of the relevant heart (HN) interneurons of the core CPG, the heart (HE) motor neurons and the heart tube segments of midbody ganglia 8 to 14. Large colored circles are cell bodies and associated input processes. Cells with similar input and output connections and function share a circle. Lines indicate cell

*Figure 1 continued*

processes, small circles indicate inhibitory chemical synapses, diodes electrical connections. The HN interneurons of ganglia 3 and 4 (HN(R/L,3) and HN(R/L,4) are part of the beat timing network and make mutual inhibitory connections. The HN(R/L,1,2) coordinating interneurons of the timing network are not illustrated for simplicity. Four pairs of premotor HN interneuron (front premotor interneurons: HN(3) and HN(4); middle premotor interneurons HN(6) and HN(7)) make inhibitory connections to all ipsilateral HE motor neurons shown here (HE(8) to HE(14)). The two bilateral heart tubes, which run the length of the animal, form the motor plant. Each individual heart segment is entrained by phasic excitatory input from its ipsilateral segmental HE motor neuron. The entire heartbeat system switches between two coordination states, left rear-to-front peristaltic/right synchronous and vice versa (peristaltic magenta, synchronous blue) about every 70 to 350 s (15 to 60 beat cycles). The two intersegmental coordination states of interneurons, motor neurons and hearts are set up by interactions between the timing network's front premotor interneurons and the middle premotor interneurons linked by the switch HN interneurons of segment 5 (HN(R/L,5)) and by direct electrical connections. On the synchronous side the switch interneuron HN(5) (ochre) bursts with the beat timing but the HN(5) on the peristaltic side is silent (greyed out). Note that the switch interneurons make bilateral connections to the middle premotor interneurons and that phasing in these CPG premotor interneurons is dominated by the single active switch interneuron. Switches (double-headed vertical arrow) in coordination state occur when the silent switch interneuron starts to burst and the bursting switch interneuron simultaneously becomes silent. The CPG switches between left synchronous/right peristaltic (top) and left peristaltic/right synchronous (bottom) states of coordination.

DOI: https://doi.org/10.7554/eLife.31123.003

The following video and figure supplement are available for figure 1:

**Figure supplement 1.** Imaging the hearts and analyzing the bilateral beat pattern of adult leeches in vivo.
DOI: https://doi.org/10.7554/eLife.31123.004

**Figure 1—video 1.** Ventral view of a flattened, intact leech illuminated from below (same animal as in *Figure 1— figure supplement 1*, anterior is to the left).
DOI: https://doi.org/10.7554/eLife.31123.005

large variability in phase across individuals at each level and show that high variability at one level is not necessarily fed forward to the next. In seeking to elucidate the sources of this population variability, we show that repetitions of a coordination state have low variability and thus contribute little. On the other hand, we show that when the same motor act is performed by bilaterally homologous neurons and muscles, variability can be as large as in the population itself depending on the level and coordination.

# Results

## Background

Medicinal leeches have two bilateral heart tubes which run the entire length of the animal (*Maranto and Calabrese, 1984a*, *1984b*; *Thompson and Stent, 1976a*). Segmental heart (HE) motor neurons innervate the hearts along their length, timing and coordinating their constrictions. The HE motor neurons are controlled by a heartbeat CPG that produces a bilaterally asymmetric activity pattern (*Calabrese, 1977*). On one side, CPG premotor interneurons fire bursts in a peristaltic rear-to-front progression and in near synchrony on the other. Motor neurons fire correspondingly leading to peristaltic and synchronous motor patterns, in turn leading to an asymmetric beat pattern of the hearts (*Wenning et al., 2004a*, *2004b*). The beat period in leeches is about 4.5 to 11 s, which translates into several thousand heartbeats per day. Embedded in this ongoing activity is the periodic alternation between the two coordinations about every 70–350 s creating episodes of coordination that span 15 to 60 beat cycles. We define a switch cycle of a given side when it has completed both coordinations, one after the other.

The core heartbeat CPG consists of 7 bilaterally paired segmental heart interneurons (HN) linked by inhibitory synapses and electrical coupling (*Figure 1*; review: *Calabrese, 2010*). Beat timing is determined by the mutually inhibitory bilateral pairs (Right/Left) of interneurons in ganglion 3 (HN(R/L,3)) and ganglion 4 (HN(R/L,4)) linked by coordinating interneurons HN(R/L,1) and HN(R/L,2), which form a beat timing network and ensure bilaterally symmetrical timing (*Calabrese, 1977*; *Calabrese, 1979*; *Masino and Calabrese, 2002a*, *2002b*, *2002c*). Output to the segmental heart motor

neurons, which occur in segments 3 to 18, is provided by inhibitory input from premotor interneurons. Each HE motor neuron makes an excitatory connection to the ipsilateral heart section in its home segment. The HN(R/L,5) interneurons switch the network. The switch interneuron on the synchronous side bursts with beat timing while the one on the peristaltic side is silent (*Calabrese, 1977*; *Gramoll et al., 1994*). Switches in coordination occur when the silent switch interneuron starts to burst and the bursting switch interneuron simultaneously becomes silent. Because the switch interneurons make bilateral connections to the middle premotor interneurons HN(6) and HN(7), phasing in these CPG premotor interneurons is dominated by the single active switch interneuron (*Figure 1*). In summary, a single CPG consisting of bilateral homologous pairs of HN interneurons produces an asymmetric premotor pattern, motor pattern, and beat pattern that episodically and periodically switches between left synchronous/right peristaltic and left peristaltic/right synchronous states of coordination (*Figure 2 A2, B2, C2*).

For this study, we focused on segments 8 to 14, because here each HE motor neuron receives input from all four ipsilateral front and middle premotor HN interneurons (HN(3), HN(4), (HN(6), HN(7)) (*Figure 1*; *Norris et al., 2007a*, *2007b*; *Thompson and Stent, 1976a*). In segments 3 to 6 and 15 to 18 the HE motor neurons receive input from additional HNs (*Norris et al., 2007a*; *Wenning et al., 2011*).

We collected data from three levels (CPG, motor neurons, and motor plant) and from the two coordinations on the two body sides resulting in 12 scenarios ($3 \times 2 \times 2 = 12$). The Project Database, illustrated in *Figure 2—figure supplement 1A–D*, was compiled since 2008 and partially reported (*Norris et al., 2011*; *Wenning et al., 2014*; *Wright and Calabrese, 2011a*), but all Bilateral Recordings and all analysis are novel.

For the sake of clarity, we report and discuss the data, and present the figures, on all three levels for one coordination (peristaltic) and for one body side (left; except when discussing bilateral variability). All data (left and right side, both coordinations) accompany the relevant Figures as source data (in table format). There were no differences in the main conclusions for the right body side. Differences in the data for synchronous vs. peristaltic coordination are pointed out and discussed. All N's reported represent the number of different animals recorded.

We focused on phase, which is a critical output characteristic of any coordinated motor program and its underlying neuronal circuitry. Phase and period do not correlate in the CPG pattern or in the motor pattern across animals in our Project Database (N = 153; data not shown) or in the beat patterns of both adult and juvenile leeches (*Wenning et al., 2004a*, *2004b*). The phase difference between the activity phases of two segments ($\Delta\phi$) is a good metric for characterizing the two coordination states (*Norris et al., 2006*; *Norris et al., 2007a*; *Norris et al., 2007b*; *Norris et al., 2011*; *Wenning et al., 2004a*; *Wenning et al., 2004b*; *Wright and Calabrese, 2011b*).

## Bilateral recordings

For the CPG pattern, we recorded from two pairs of heart interneurons – HN(L/R,4) and HN(L/R,7) – (*Figure 2A1*), for the motor pattern we recorded from two pairs of motor neurons – HE(L/R,8) and the HE(L/R,12) – (*Figure 2B1*), and for the beat pattern, we extracted the digitized optical signals for heart (L/R,8) and heart (L/R,12) (*Wenning et al., 2014*; *Figure 2C1*).

All sample recordings (*Figure 2A2, B2, C2*) start with the left side in peristaltic coordination (rear-to-front delay) and the right in synchronous coordination. In the CPG, the HN(L,7) interneuron bursts lead those of the HN(L,4) interneuron, while the HN(R,4) interneuron slightly leads the HN(R,7) interneuron. Similarly, the HE(L,12) motor neuron bursts lead those of the HE(L,8) motor neuron while the HE(R,8) motor neuron bursts slightly lead those of the HE(R,12) motor neuron. Finally, heart (L,12) starts to constrict before heart (L,8) while the right side is in synchronous coordination with the hearts (R,12) and (R,8) constricting almost synchronously. *Figure 1—figure supplement 1* and *Figure 1—video 1* show the beat pattern for this preparation. In all recordings, the two sides switch coordinations simultaneously to left synchronous/right peristaltic (*Figure 2A2, B2, C2*).

The triangles of *Figure 2A3, B3* and *C3* show the average $\Delta\phi$ between the left front and rear segment of one switch cycle, i.e. between the HN(L,4) and HN(L,7) interneurons (*Figure 2A3*), between the HE(L,8) and HE(L,12) motor neurons (*Figure 2B3*), and between heart (L,8) and heart (L,12) (*Figure 2C3*). The circular phase plots of *Figure 2A4, B4, C4* illustrate the cycle-to-cycle variability for all cycles in peristaltic coordination.

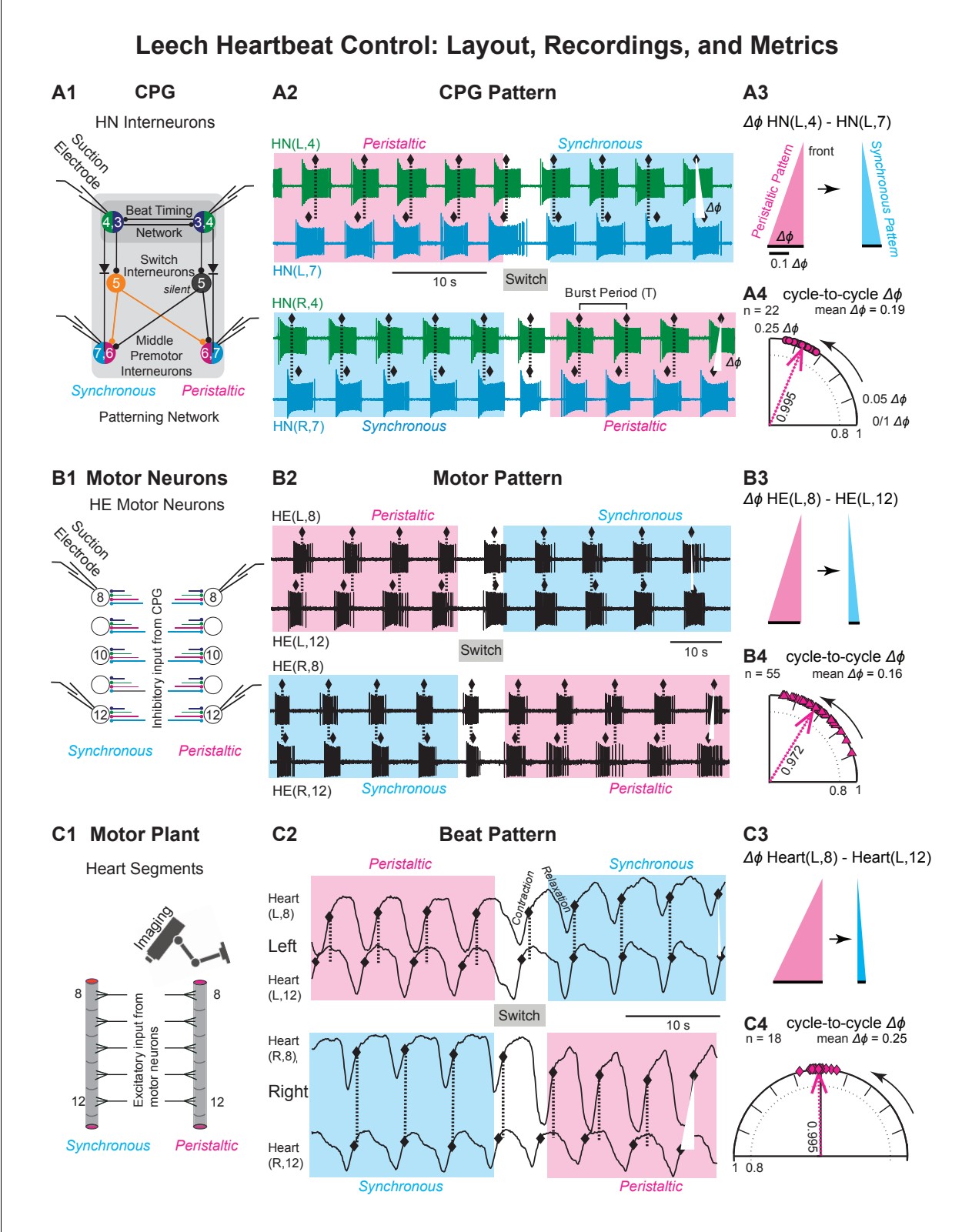

**Figure 2.** Recordings of the CPG pattern, the motor pattern, and the beat pattern. (**A1, B1, C1**) Recording sites and methods for all levels of the heartbeat control system. (**A2, B2, C2**) Recordings for all levels of the heartbeat control system. (**A3, B3, C3**) The base of the triangle represents the intersegmental phase differences ($\Delta\phi$) between the front segment and the rear segment in the two coordinations (left side). (**A4, B4, C4**) Circular phase plots illustrate the cycle-to-cycle variability in peristaltic coordination (left side). The vector length (value next to each vector) was used to calculate the

*Figure 2 continued on next page*

*Figure 2 continued*

angular variance ($s^2 = 2(1 \ r)$). Longer vectors indicate less variance. (**A1**) Suction electrodes were placed on the left and right HN(4) and HN(7) interneurons. (**A2**) Simultaneous extracellular recordings from these four premotor HN interneurons across a switch in coordination state (top: left heart interneurons; HN(L,4) and HN(L,7); below: right heart interneurons (HN(R,4) and HN(R,7)). Phase marker (♦) is the middle spike in each burst. Dashed lines and white triangles aid in assessing the phase differences between the two interneurons in the two coordinations. Initially, the left side is peristaltic (magenta shaded box) and the right side is synchronous (light blue shaded box). State switches midway. (**A3**) Colored triangles illustrate the average intersegmental $\Delta\phi$ (black bars) between the ipsilateral pair of HN interneurons in the peristaltic (magenta) and synchronous (light blue) coordinations of the recording of (**A2**). (**A4**) The circular phase plot shows the cycle-to-cycle variability in peristaltic coordination for the recording of (**A2**). Each circle represents the $\Delta\phi$ between the two ipsilateral HN interneurons of one burst cycle (n = 22 bursts, mean $\Delta\phi$ = 0.19). (**B1**) Suction electrodes were placed on the left and right HE(8) and HN(12) motor neurons. (**B2**) Simultaneous extracellular recordings from these four heart motor neurons across a switch in coordination state (top, left motor neurons HE(L,8) and HE(L,12); below, right motor neurons (HE(R,8) and HE(R,12)). Phase marker (♦) is the middle spike in each burst. Labeling as in (**A2**). Initially, the left side is peristaltic and the right side is synchronous. State switches midway. (**B3**) Labels as in (**A3**). (**B4**) The circular phase plot shows the cycle-to-cycle variability in peristaltic coordination for the recording of (**B2**). Each triangle represents the $\Delta\phi$ between the two ipsilateral heart motor neurons for one burst cycle (n = 55 bursts, mean $\Delta\phi$ = 0.16). Labeling as in (**A4**). Phase scale bars as in (**A3**). (**C1**) Video imaging of intact animals yielded optical signals to extract the constriction/relaxation cycles for both hearts in multiple segments. (**C2**) The beat cycles of two bilateral pairs of heart segments are shown across a switch in coordination state (top: Heart (L,8) and Heart (L,12); below: Heart (R,8) and Heart (R,12)). Phase marker (♦) is the maximum rate of rise (MRR) during the constriction (*Wenning et al., 2014*). Initially, the left side is peristaltic and the right side is synchronous. State switches midway. (**C3**) Labels as in (**A3**). (**C4**) The circular phase plot shows the cycle-to-cycle variability in peristaltic coordination for the video recording of (**C2**). Each diamond represents the $\Delta\phi$ between the two ipsilateral heart segments for one beat cycle (n = 18 beats, mean $\Delta\phi$ = 0.25). Labeling as in (**A4**) Phase scale bars as in (**A3**). Data from the animal shown in *Figure 1—video 1*. Animal Groups: Bilateral Recordings (*Figure 2—figure supplement 1C*) and Intact Animal Database (*Figure 2—figure supplement 1D*).

DOI: https://doi.org/10.7554/eLife.31123.006

The following figure supplement is available for figure 2:

**Figure supplement 1.** Project Database and animals used in this study.

DOI: https://doi.org/10.7554/eLife.31123.007

## Variability within animals: Cycle-to-cycle variance

Each bout of behavior, either peristaltic or synchronous, has between 15 and 60 neuronal bursts (CPG pattern, motor pattern) and rhythmic constrictions (beat pattern). How variable are the patterns across the bursts or beats within one bout of behavior? *Figure 2* illustrates the individual $\Delta\phi$ between two segments of these individual bursts and constrictions, referred to as cycle-to-cycle variability, and their variance, for a single preparation for the CPG pattern (A4), the motor pattern (B4), and for the beat pattern (C4). *Figure 3A* shows the variances for all preparations used in this study for two subsequent switch cycles. Cycle-to-cycle variances of the output patterns were equally low on the two sides, in both coordinations, and on all levels (*Figure 3—source data 1*). But what does 'low' mean? We reasoned that the timing network had the lowest variabilities in the heartbeat system. In the same 26 animals where we recorded the CPG pattern (*Figure 2—figure supplement 1D*) we calculated the cycle-to-cycle variances of the phase difference between the two HN(4) interneurons, which are part of the timing network and which form a half-center oscillator with strong mutual, inhibitory connections (*Figure 1*; *Calabrese, 1977*; *Hill et al., 2001*; *Sorensen et al., 2004*). The average cycle-to-cycle variance of the $\Delta\phi$ between the two HN(4) interneurons was similar to the average cycle-to-cycle variances of the $\Delta\phi$ between two premotor interneurons, two motor neurons, and two heart segments (*Figure 3A*). In these same recordings, we assessed period variability and found the coefficient of variation to be on average of less than 5% (*Figure 3B*).

The low cycle-to-cycle variability indicates a highly coordinated and precise motor system and allowed us to use the average $\Delta\phi$ of a given coordination in an individual to assess the population, repetition, and bilateral variability.

## Variability across animals: population variance

We plotted the average $\Delta\phi$ between two segments as detailed above in circular phase plots using the animals of this study (*Figure 2—figure supplement 1D* and *Figure 4*), for the Project Database (*Figure 2—figure supplement 1A* and *Figure 5A*), and for the Simultaneous Recordings from the HN and HE neurons (*Figure 2—figure supplement 1B* and *Figure 5B*).

We had shown previously that the average $\Delta\phi$ differences between the front and middle premotor HN interneurons (HN(4) and HN(7)) were larger than those between the HE(8) and the HE(12) motor neurons (*Wright and Calabrese, 2011a*). We obtained the same results in three data sets: (1)

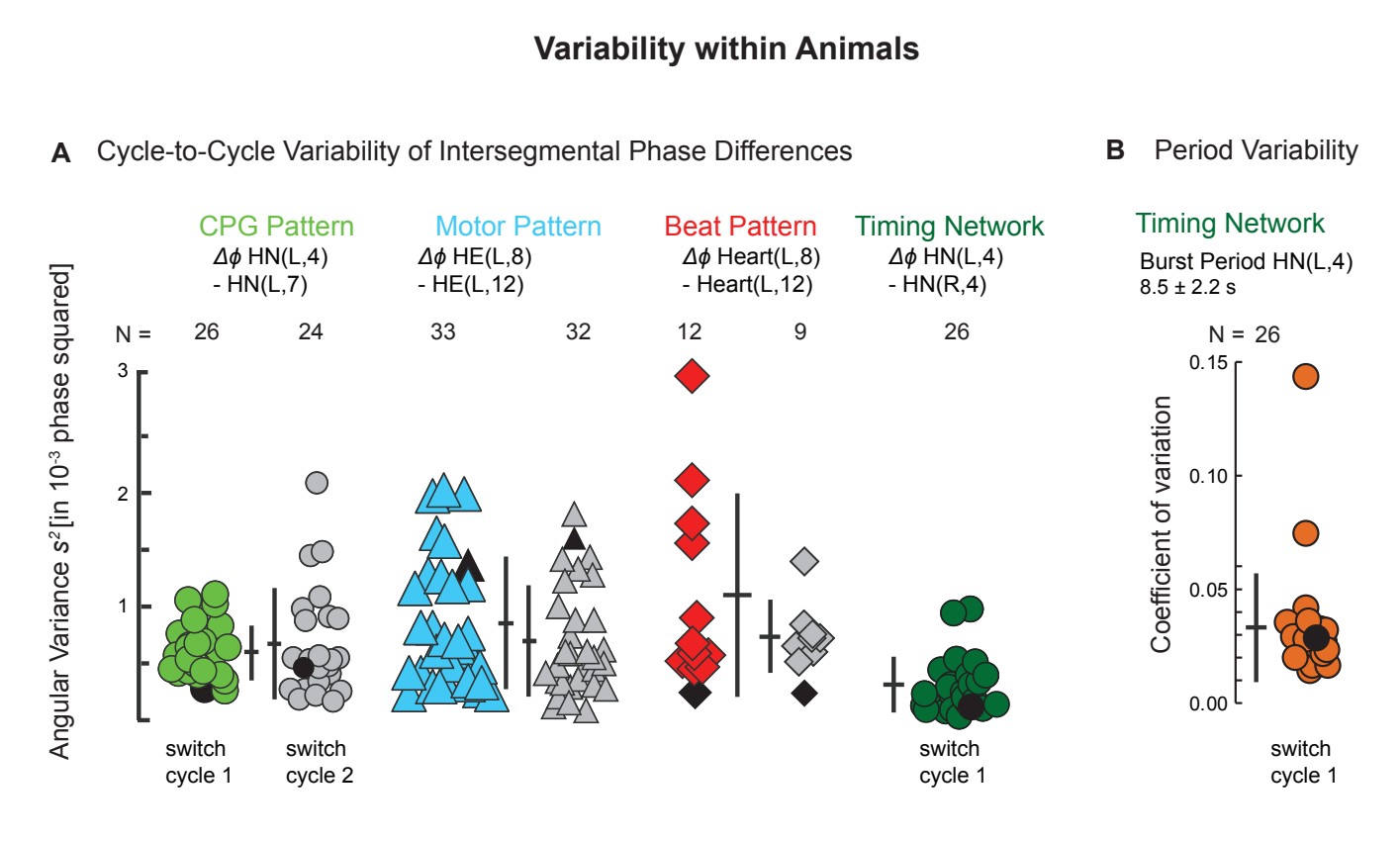

**Figure 3.** Variability within animals: cycle-to-cycle and period variability across individuals. (**A**) The angular variance $s^2$ of intersegmental phase differences for all bursts (CPG, motor neurons) and beats (hearts) are plotted for two subsequent switch cycles for the CPG pattern (green circles), the motor pattern (aqua triangles), and the beat pattern (red diamonds) within side (left) and coordination (peristaltic). Each symbol represents one preparation. The number of preparations is indicated for each level. Colored symbols: switch cycle 1; grey symbols: switch cycle 2. The angular variance is also shown for one switch cycle for the timing network of the CPG (side-to-side phase difference between the left and the right HN(4) interneurons; dark green circles). Note the low variability on all levels. Number of cycles per switch cycle (n's) were 7 to 58 (CPG), 14 to 68 (Motor Pattern), and 7 to 31 (Beat Pattern). (**B**) Period variability of the timing network is shown as the coefficient of variation for the 26 preparations of switch cycle 1. Black symbols represent the preparations shown in **Figure 2**. Means ± SD are shown next to each group as horizontal and vertical bars, respectively. Animal Groups: Bilateral Recordings (**Figure 2—figure supplement 1C**) and Intact Animal Database (**Figure 2—figure supplement 1D**).

DOI: https://doi.org/10.7554/eLife.31123.008

The following source data is available for figure 3:

**Source data 1.** Cycle-to-Cycle variances for the CPG pattern, the motor pattern, and the beat pattern for both coordinations, peristaltic and synchronous, and for both sides.

DOI: https://doi.org/10.7554/eLife.31123.009

Bilateral Recordings, **Figure 2—figure supplement 1C** and **Figure 4B**: unpaired t-test, p<0.001; (2) Project Database, **Figure 2—figure supplement 1A** and **Figure 5A**: unpaired t-test, p<0.001; (3) Simultaneous Recordings of the CPG pattern and the motor pattern, **Figure 2—figure supplement 1B** and **Figure 5C**: paired t-test, p<0.001. Finally, the intersegmental $\Delta\phi$ of the CPG pattern also exceeded that of the motor pattern (paired t-test, p<0.001) in the nine simultaneous Bilateral Recordings of the CPG pattern and the motor pattern (**Figure 2—figure supplement 1C**; both coordinations, both sides; data not shown).

Next, we determined the angular variances of these intersegmental phase differences for the animals in which we made bilateral recordings (**Figure 2—figure supplement 1C**), calculated the population variance for each level, and determined the confidence interval for each level with bootstrapping (10,000 times with replacement) (significance level: 0.05; **Figure 4C**). The population variances of these intersegmental $\Delta\phi$ were substantial (among the largest variances determined in this study) indicating considerable variability in the population. The smaller population of bilateral

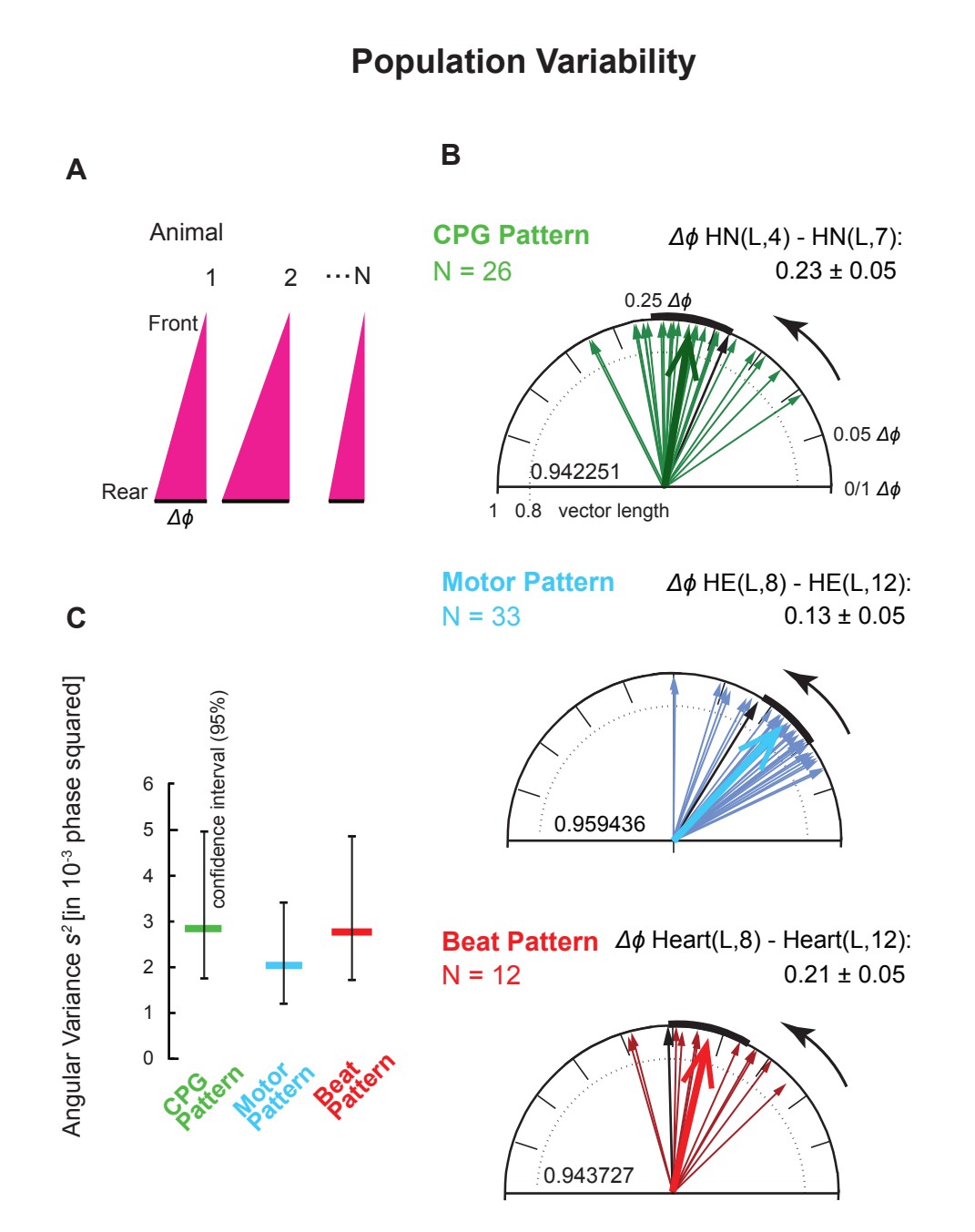

**Figure 4.** Intersegmental phase variability across animals. (**A**) Population variances were calculated using the average intersegmental $\Delta\phi$ of all bursts or beats in a single switch cycle per animal (peristaltic, left side). (**B**) Circular plots show the population variability for each level (color code *Figure 3*). Each thin arrow represents the average intersegmental $\Delta\phi$ of one preparation and its length represents the angular variance of the cycle-to-cycle variability of that animal. Each thick arrow represents the average intersegmental $\Delta\phi$ across preparations (values indicated for each level), its length is inversely proportional to the angular variance (vector lengths given inside the circles), and the black arc is the angular standard deviation. Note that the average intersegmental $\Delta\phi$ is smaller for the motor pattern than for the CPG pattern. The number of preparations (N's) and the mean (±SD) intersegmental $\Delta\phi$ is indicated for each level. Black arrows are the vectors of the preparations shown in *Figure 1* (their cycle-to-cycle variances are shown as black symbols in *Figure 4*). (**C**) Angular variances are similar for all patterns (colored horizontal bars). The confidence intervals (95%) of 10,000 bootstrapped populations overlap across levels (vertical

*Figure 4 continued*

lines). Animal Groups: Bilateral Recordings (*Figure 2—figure supplement 1C*) and Intact Animal Database (*Figure 2—figure supplement 1D*).

DOI: https://doi.org/10.7554/eLife.31123.010

The following source data and figure supplement are available for figure 4:

**Source data 1.** Intersegmental phase differences and population variances for the CPG pattern, the motor pattern, and the beat pattern for both coordinations, peristaltic and synchronous, and for both sides.

DOI: https://doi.org/10.7554/eLife.31123.012

**Figure supplement 1.** Angular variances for three motor patterns (peristaltic; left side).

DOI: https://doi.org/10.7554/eLife.31123.011

recordings reported here (*Figure 2—figure supplement 1C*) nevertheless represents the larger populations (*Figure 2—figure supplement 1A,B*) reasonably well, because their confidence intervals overlap extensively (compare *Figure 4* and *Figure 5*).

We had previously shown that at none of the network levels was intersegmental $\Delta\phi$ correlated with the cycle period (*Norris et al., 2006*; *Wenning et al., 2004a*; *Wenning et al., 2004b*). We corroborated these results for the CPG pattern and the motor pattern using our Project Database (*Figure 2—figure supplement 1A*) and found no correlation over a cycle period range of 4 to 13 s for the 129 HN interneurons and of 5 to 11 s for the 83 heart motor neurons (data not shown).

## Variability across animals: comparison across levels

The variance of the motor pattern in the bilateral recordings of *Figure 4C* appears to be lower than that of the CPG or motor plant although the confidence intervals overlap. To clarify whether the motor pattern does indeed have a lower variance than the CPG pattern, we used our larger databases. We calculated the angular variances using the entire Project Database (*Figure 2—figure supplement 1A*) and found that variances were higher in the CPG pattern than in the motor pattern (0.0038, CPG pattern; 0.0020, motor pattern) with no overlap in the bootstrapped 95% confidence intervals (*Figure 5B*). To eliminate the possibility that this difference results from CPG and motor pattern recordings being made in different preparations, we calculated the angular variances from simultaneous (mostly unilateral) recordings of the CPG pattern and the motor pattern (*Figure 2—figure supplement 1B*). We found that variances were also higher in the CPG pattern than in the motor pattern (0.0043, CPG pattern, 0.0018 motor pattern), again with no overlap in the bootstrapped 95% confidence intervals (*Figure 5D*).

Two aspects of the CPG output determine the motor pattern for the HE motor neurons of segments 8 to 14: the $\Delta\phi$ of the premotor interneurons of the CPG and their synaptic strength (*Wright and Calabrese, 2011a*, *2011b*). For this study, we quantified $\Delta\phi$ and synaptic strength for two of the four pairs of the premotor interneurons on both sides simultaneously (*Figure 2A1*). These factors combine with the intrinsic properties of the HE motor neurons and their electrical coupling between bilateral homologs to determine when an individual HE motor neuron fires in a heartbeat cycle (*Shafer and Calabrese, 1981*; *Wright and Calabrese, 2011b*). While the phase difference of the premotor HN interneurons is the same in all the segments considered here, the synaptic strength of their connections to motor neurons progressively changes across segments although there is considerable individual variability (*Norris et al., 2006*, *2007a*; *Norris et al., 2011*; *Wright and Calabrese, 2011a*, *2011b*). In peristaltic coordination, the phase progression of the premotor bursting pattern determines the maximal phase *range*, and segment-specific synaptic strength pattern, intrinsic properties and coupling determines the phase *realized* between two ipsilateral motor neurons (*Wright and Calabrese, 2011b*). The $\Delta\phi$ that the motor neurons achieve is a portion of the $\Delta\phi$ of the premotor interneurons of the CPG depending on the number of segments considered (*Figure 4* and *Figure 5*). Therefore, because the $\Delta\phi$ of the motor pattern (HE(8) to HE(12)) is significantly smaller than that of the CPG pattern (*Figure 4—source data 1*), it expresses less of the CPG's variability. Under this hypothesis, as more or less of the CPG's $\Delta\phi$ is expressed then more or less of its variability is expressed. We found that this was the case. In some of our bilateral HE recordings we recorded the HE(14) (N = 15) or the HE(10) (N = 9) along with the HE(8). Indeed, variances increased as the motor pattern's $\Delta\phi$ approached that of the CPG as more segments intervened between recorded motor neurons (*Figure 4—figure supplement 1*).

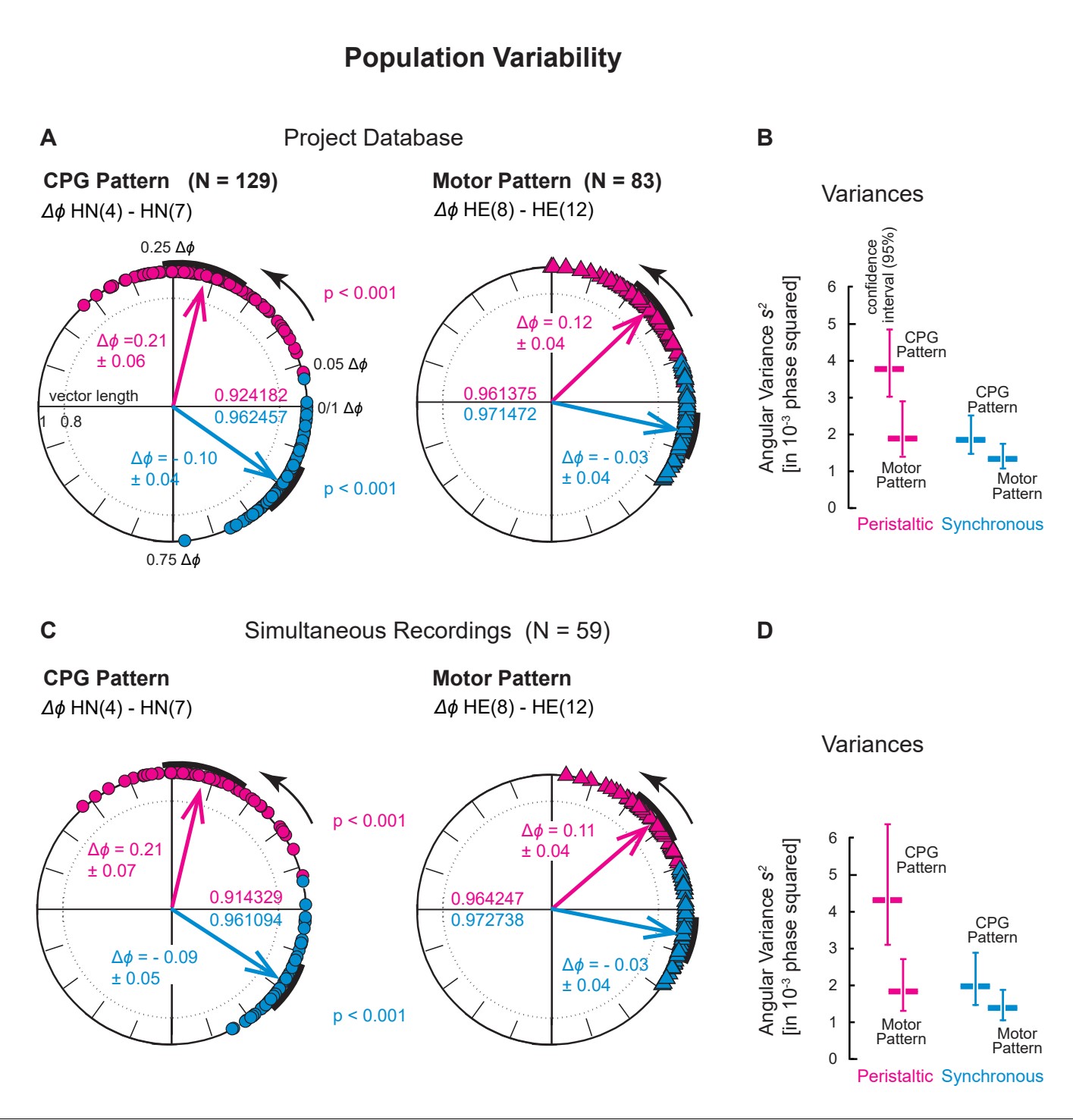

**Figure 5.** Population variability across two larger populations. Population variances for the Project Database (Top) and for the Simultaneous Recordings from the HN interneurons and HE motor neurons (Bottom). (**A**) Variances across all preparations in the Project Database (*Figure 2—figure supplement 1A*). Each symbol on the circular phase plots represents the average intersegmental $\Delta\phi$ of one preparation in one switch cycle. The number of preparations and the mean intersegmental $\Delta\phi$ (±SD) is indicated for the CPG pattern (N = 129; circles) and of the motor pattern (N = 83, triangles) for peristaltic (magenta) and synchronous (blue) coordination. Thick arrows represent the average intersegmental $\Delta\phi$ across preparations (values indicated for each level), their length the angular variance (values inside each circle), and the black arcs the angular standard deviation. Note that the intersegmental $\Delta\phi$ of the CPG pattern is larger than that of the motor pattern (peristaltic: p<0.001, synchronous: p<0.001, unpaired t-test). (**B**) Angular variances and the confidence intervals of 10,000 bootstrapped populations (vertical lines) for the CPG pattern and the motor pattern. Note that

*Figure 5 continued on next page*

*Figure 5 continued*

confidence intervals (95%) do not overlap in the peristaltic coordination. (**C**) Variances across all Simultaneous Recordings (*Figure 2—figure supplement 1B*). Each symbol on the circular phase plot represents the average $\Delta\phi$ of a preparation in which HN interneurons and HE motor neurons were simultaneously recorded. Layout, symbols and colors as in (**A**). Note that the average intersegmental $\Delta\phi$ of the CPG pattern is larger than that of the motor pattern (peristaltic: p<0.001, synchronous: p<0.001, paired t-test). (**D**) Angular variances and their confidence intervals are shown for the CPG pattern and the motor pattern. Note that confidence intervals (95%) do not overlap in peristaltic coordination. Color code and labels as in (**B**).

DOI: https://doi.org/10.7554/eLife.31123.013

## Repetition variance

While ongoing, leech heartbeat is episodic; the CPG, motor neurons, the hearts alternate between two coordination states at regular intervals (*Figure 2*; *Calabrese, 2010*). How similar is the same coordination when repeated on the same side a few minutes later by the exact same neurons? We assessed this repetition variability at all levels using the Bilateral Recordings and the Intact Animal Database where we had imaged the beat pattern on both sides (*Figure 2—figure supplement 1C, D*). We subtracted the average intersegmental $\Delta\phi$ of one switch cycle from that of another, subsequent switch cycle ($\Delta\Delta\phi = \Delta\phi_1 - \Delta\phi_2$; *Figure 6A*) and calculated the variance of that distribution. This difference between the two intersegmental phase differences ($\Delta\Delta\phi$) is 0 when the two switch cycles have identical intersegmental phase differences ($\Delta\phi s$). The circular phase plots of *Figure 6B* show the $\Delta\Delta\phi$ between two consecutive switch cycles across animals for all levels, and show that, indeed, the average phase difference is near 0.

We found that consecutive bouts of the same pattern differed in half of the preparations (CPG pattern: 10 of 24; motor pattern: 17 of 32; beat pattern: 5 of 9; unpaired t-tests; *Figure 6—source data 1*). To evaluate the difference between repetitions, we compared these $\Delta\Delta\phi$ to the average intersegmental phase difference $\Delta\phi$, and found the $\Delta\Delta\phi$ to be an order of magnitude smaller ($\Delta\Delta\phi$ vs D$f$ : 0.028 vs 0.23, CPG pattern; 0.021 vs 0.13, motor pattern; 0.023 vs 0.21, beat pattern). To determine how the repetition variance compared with that across animals, we calculated the variance of scrambled pairs of switch cycles 1 and 2 using all preparations in the dataset and repeated this procedure 10,000 times. At all levels, the repetition variance in the original population was much smaller than in the scrambled populations (*Figure 6C*; p<0.005 for all levels). The differences between repetitions that we did find can be attributed not to their large size but to the overall low cycle-to-cycle variances (*Figure 3*).

Repetition variances were equally low on the two sides and in both coordinations. The one exception which had a high repetition variance (right heart, synchronous) is most likely due to an outlier in this small sample. On all levels, across coordinations and sides, repetition variances were significantly lower than those in the population (*Figure 6* and *Figure 6—source data 1*). Our results show that the repetition variances were small and suggest that they do not contribute substantially to the population variance.

## Bilateral variance

The leech heartbeat system is composed of bilaterally homologous elements (interneurons, motor neurons, and hearts; *Figure 1* and *Figure 2*) which allowed us to assess how similar the same coordination is when executed by the genetically identical contralateral homologs of the same neurons and muscles. We assessed this bilateral variability at all levels using the Bilateral Recordings and the animals where we had imaged the beat pattern on both sides (*Figure 2—figure supplement 1C,D*). We subtracted the average intersegmental $\Delta\phi$ of one side from that of the other side (within coordination and level) ($\Delta\Delta\phi = \Delta\phi_R - \Delta\phi_L$; *Figure 7A*) and calculated the variance of that distribution. This difference between the two intersegmental phase differences ($\Delta\Delta\phi$) is 0 when the two sides have identical intersegmental phase differences($\Delta\phi s$). The circular phase plots of *Figure 7B* show the $\Delta\Delta\phi$ between the two sides across animals for each level.

The mean $\Delta\Delta\phi$ at each level was near zero and evenly distributed, eliminating the possibility of a dominant form of handedness in the heartbeat system. Bilateral variances at each level were substantial (*Figure 7C*). We found that the mean of the absolute values of the $\Delta\Delta\phi s$ ($|\Delta\Delta\phi|$) between sides were about 2–3 times larger than the $|\Delta\Delta\phi|s$ between repetitions (CPG pattern: 0.06 vs 0.03; motor pattern: 0.04 vs 0.02; beat pattern: 0.06 vs 0.02; compare *Figure 6—source data 1* and

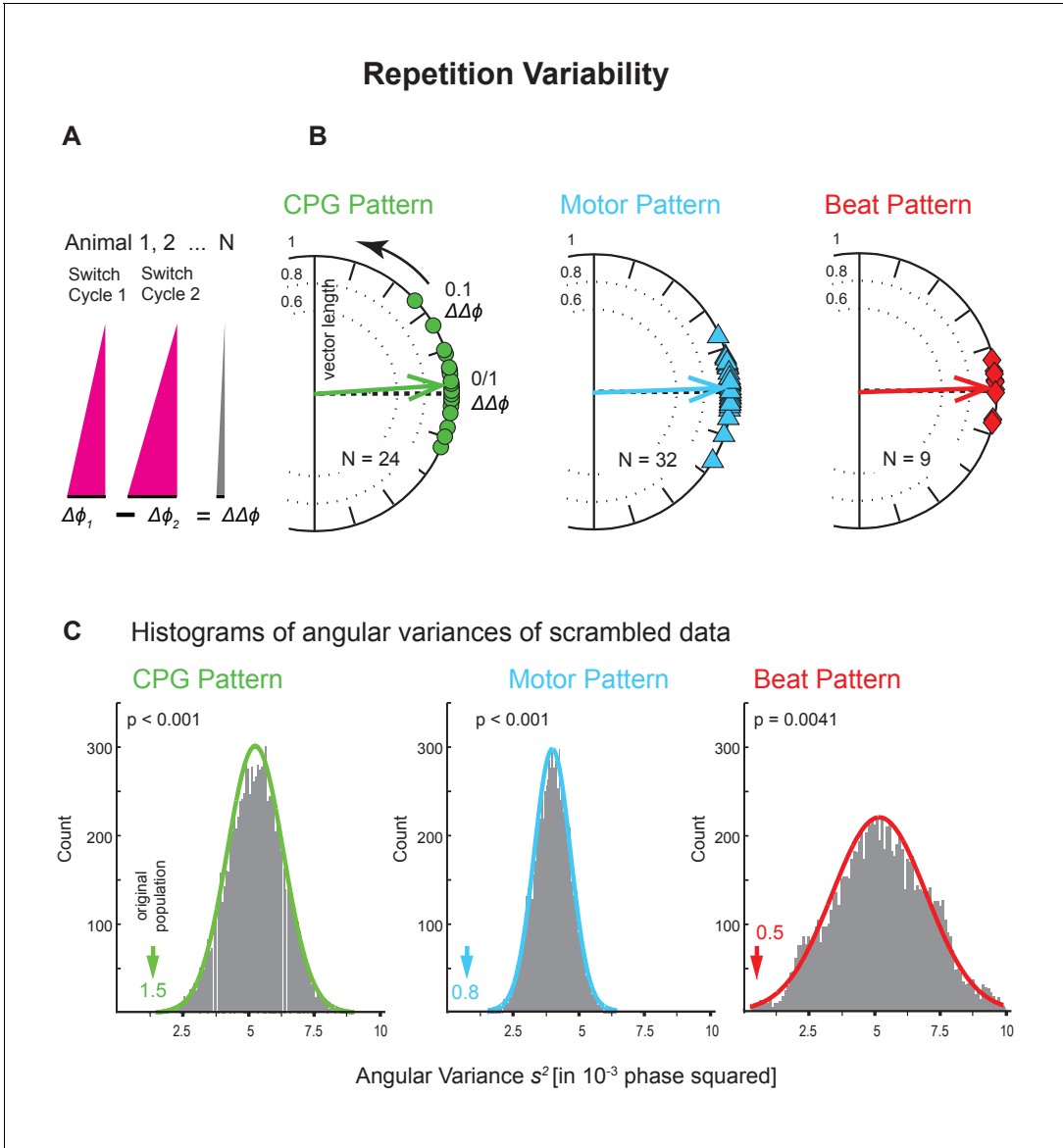

**Figure 6.** Repetition variability across levels and animals. (A) The average intersegmental $\Delta\phi$ of two subsequent switch cycles were subtracted from each other ($\Delta\phi_1$ - $\Delta\phi_2$). This difference of phase differences ($\Delta\Delta\phi$) becomes 0 if $\Delta\phi_1$ and $\Delta\phi_2$ are identical. (B) Circular phase plots show $\Delta\Delta\phi$ for all preparations for all levels colors as in *Figure 4*. Each symbol on the circular phase plots represents the $\Delta\Delta\phi$ of one preparation. Thick arrows show the average $\Delta\Delta\phi$ and their length the angular variance. Note that vector phases are close to 0. (C) Histograms of the angular variances of 10,000 populations where the average intersegmental $\Delta\phi$ of switch cycle two from one animal had been randomly subtracted from the average intersegmental $\Delta\phi$ of switch cycle one from another animal (scrambling). Colored arrows point to the variance of the original population (value next to each arrow). The p values for each histogram were calculated from the z score of the normal distribution (colored lines) of the scrambled populations. Note that on each level scrambling resulted in significantly higher variances than that of the original population. Data within side (left) and coordination (peristaltic). Animal Groups: Bilateral Recordings (*Figure 2—figure supplement 1C*) and Intact Animal Database (*Figure 2—figure supplement 1D*).
DOI: https://doi.org/10.7554/eLife.31123.014

The following source data is available for figure 6:

**Source data 1.** Repetition variances of the CPG pattern, the motor pattern, and the beat pattern for both coordinations and for both sides.
DOI: https://doi.org/10.7554/eLife.31123.015

*Figure 7—source data 1*). Hence bilateral variances at each level were 3-4fold higher than the repetition variances (in $10^{-3}$ phase squared: CPG pattern, 5.1 vs 1.5; motor pattern, 2.1 vs 0.8; beat pattern, 4.5 vs 0.5; compare *Figure 6—source data 1* and *Figure 7—source data 1*).

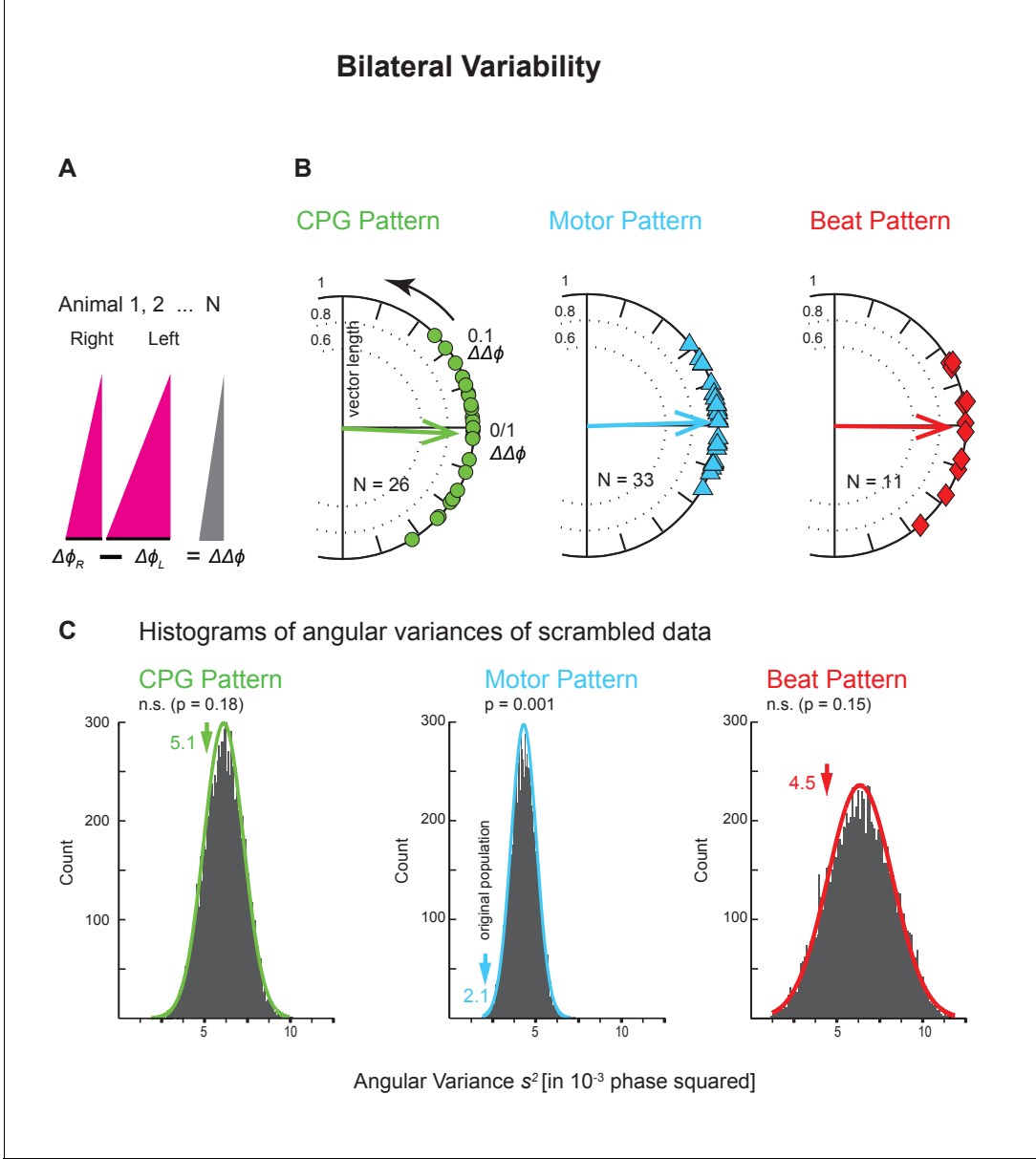

**Figure 7.** Bilateral variability across levels and animals. (**A**) The average intersegmental $\Delta\phi$ between two segments on the right and on the left body side were subtracted from each other ($\Delta\phi_L$ - $\Delta\phi_R$) (peristaltic). This difference of phase differences ($\Delta\Delta\phi$) becomes 0 if phase differences are identical on the two sides. (**B**) Circular phase plots show $\Delta\Delta\phi$ for all preparations for all levels (colors as in *Figure 6*). Each symbol on the circular phase plots represents the $\Delta\Delta\phi$ of one preparation. Thick arrows show the average $\Delta\Delta\phi$ and their length the angular variance. Note that vector phases are close to 0. (**C**) Histograms of the angular variances of 10,000 populations where the intersegmental $\Delta\phi$ on the right side of one animals had been randomly subtracted from the left side from another animal (scrambling). Colored arrows represent the variance of the original population (value next to each arrow). Note that the variances of the CPG pattern and the beat pattern of the original population do not differ from the scrambled populations, while in the motor pattern the variance from the original population is significantly lower than the variances of the scrambled population. The p values for each histogram were calculated from the z score of the normal distribution (colored line) of the scrambled populations. Layout as in *Figure 6*. Animal Groups: Bilateral Recordings (*Figure 2—figure supplement 1C*) and Intact Animal Database (*Figure 2—figure supplement 1D*).
DOI: https://doi.org/10.7554/eLife.31123.016

The following source data and figure supplement are available for figure 7:

**Source data 1.** Bilateral variances of the CPG pattern, the motor pattern, and the beat pattern for both coordinations.
DOI: https://doi.org/10.7554/eLife.31123.018

**Figure supplement 1.** Intersegmental phase variability within one switch cycle across animals.
DOI: https://doi.org/10.7554/eLife.31123.017

Moreover, we found that the bouts of the same coordination on the two sides differed in most cases (CPG pattern: 20 of 26; motor pattern: 29 of 33; beat pattern: 8 of 11; unpaired t-tests; *Figure 7—source data 1*), yet, the average phase difference is similar on the two sides (*Figure 7—figure supplement 1*).

To determine how the bilateral variance compared with the population variance, we calculated the variances of scrambled pairs (one from the left, one from the right side) using all preparations in the dataset and repeated this procedure 10,000 times. In the CPG pattern and in the beat pattern, the bilateral variance in our data set was not significantly different from the scrambled populations (*Figure 7C*; p=0.18 and p=0.15, respectively). In the motor pattern, however, the bilateral variance was significantly smaller than in the scrambled populations (p<0.001). We obtained the same result when using the HE(8) to HE(14) motor pattern where population variance was higher (*Figure 4—figure supplement 1*; plot not shown). In synchronous coordination, on all levels of the network, bilateral variances were lower in the original population than those of the scrambled data (*Figure 7C*). These results suggest that at least for the CPG and the motor plant when in peristaltic coordination, differences between homologous elements, as reflected in the bilateral variances, may contribute significantly to the population variance. Across levels, intersegmental phase differences on one side do not correlate with those on the other side (data not shown).

## Variability of synaptic strengths in the heart motor neurons

The CPG network distributes its output over an ensemble of motor neurons in a stereotyped pattern of synaptic connections (*Calabrese, 1977*; *Thompson and Stent, 1976a*). The HN(4) and HN(7) interneurons we recorded for this study make connections to all motor neuron pairs of segments 8 to 18 (*Calabrese, 1977*; *Shafer and Calabrese, 1981*; *Thompson and Stent, 1976b*) (*Figure 1*). The synaptic strengths of the individual premotor HN interneurons have distinct average segmental profiles but vary across animals (*Norris et al., 2007a*, *2007b*, *2011*). For example, on average, synaptic strength of the HN(4) interneuron is highest in the HE(8) motor neuron and weakens towards more posterior heart motor neurons while the synaptic strength of the HN(7) interneuron is highest between the HE(10) and HE(14) motor neurons. Synaptic strengths do not change with changes in coordination states (*Norris et al., 2007a*).

*Figure 8* shows the connection strength of the premotor heart interneurons to several ipsilateral motor neurons on both sides using a subgroup of the bilateral recordings from the HN interneurons ('Synaptics', *Figure 2—figure supplement 1C*). Comparing the two sides let us assess variabilities emerging during development in bilaterally homologous neurons. Specifically, we voltage-clamped three pairs of heart motor neurons (HE(R/L,8), HE(R/L,10), and HE(R/L,12)), one after the other, while simultaneously recording extracellularly from two pairs of premotor interneurons (HN(R/L,4) and HN(R/L,7)). Using spike-triggered averaging (described in *Norris et al., 2007a*), we determined the synaptic strength from the HN interneurons to their ipsilateral motor neurons in at least 10 bursts per interneuron/motor neuron pair. *Figure 8A* shows the synaptic strength from the HN(4) and HN(7) interneurons to each of the three ipsilateral heart motor neurons. As detailed above, synaptic strength of the HN(4) interneuron tends to be higher in the HE(8) than in the HE(12) motor neuron while synaptic strength of the HN(7) interneuron tends to be higher in the HE(12) than in the HE(8) motor neuron, and synaptic strengths tend to be about equal in the HE(10). Synaptic strengths on the two sides vary and are not identical – but how similar are they?

To compensate for differences in the quality of voltage-clamp recording of individual neurons we computed the proportion of the synaptic strength due to the HN(4) premotor interneuron on each side for each motor neuron (*Figure 8B*). Right and left body side did not differ (paired t-test; p=0.23 HE(8), p=0.91 HE(10), p=0.99 HE(12)), indicating no systematic bilateral bias. The mean p values after randomly combining the two sides 10 000 times were HE(8): 0.25, HE(10): 0.89, and HE(12): 0.98 (*Figure 8B*). The results show that despite a lot of variation in individual strength (nS; *Figure 8A*), the two sides do not differ in proportional synaptic strength (*Figure 8B*) indicating that bilateral variability does not seem to contribute to the population variability in synaptic strength. In the motoneuronal network of *Drosophila* larvae motor neurons of the same cell type contacting a common interneuron can have quite different numbers of synapses but whether the number of synapses correlate with synaptic strength is unknown (*Couton et al., 2015*).

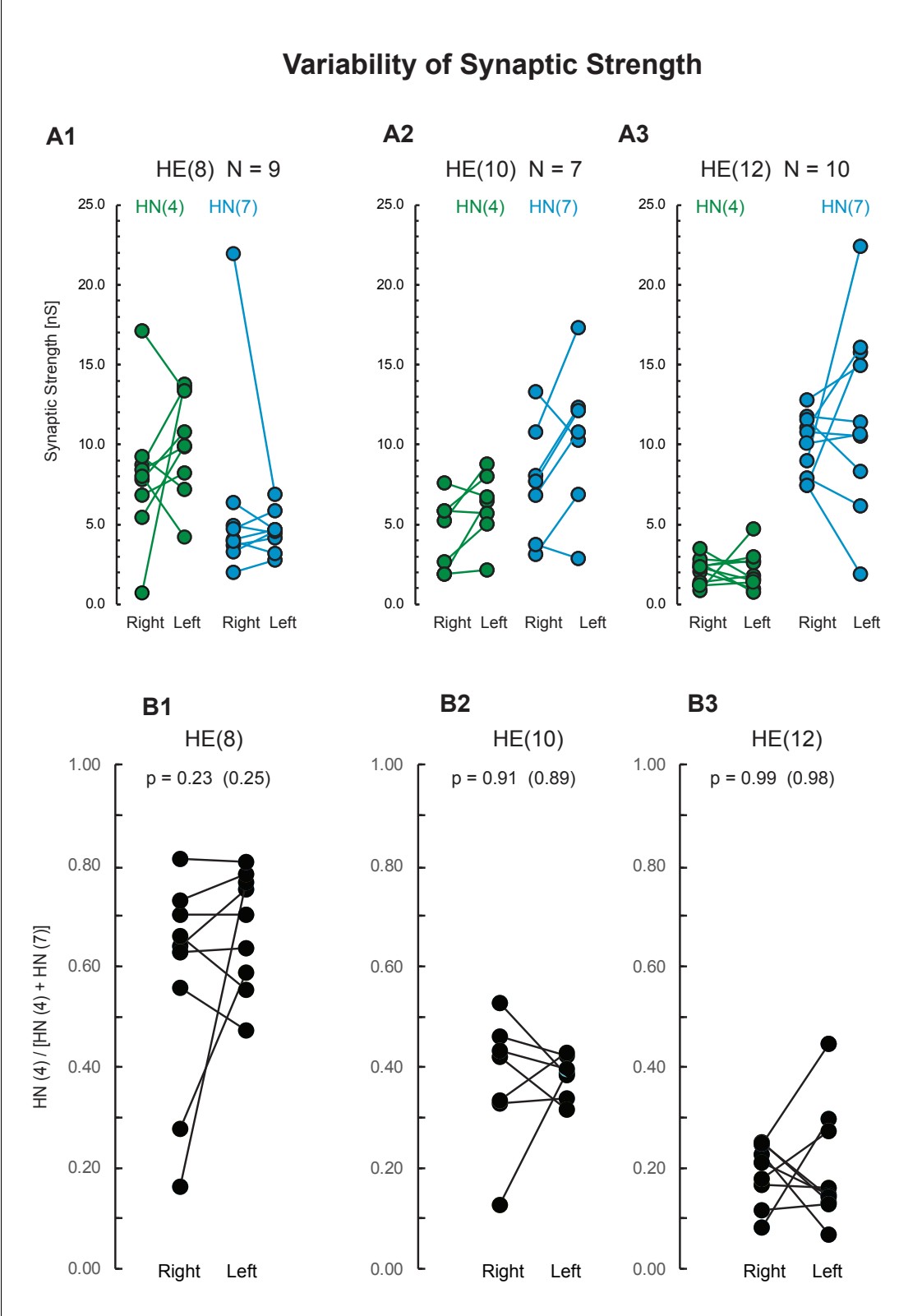

**Figure 8.** Variability of synaptic strength across animals. (**A**) Each data pair represents the right and the left synaptic strength (in nS) from premotor interneurons HN(4) (green) and HN(7) (blue) to the ipsilateral heart motor neurons HE(8) (**A1**), HE(10) (**A2**), and HE(12) (**A3**) in the same animal. Synaptic strength does not differ between sides (paired t-test; p values on plot). (**B**) Proportion of the total HN(4) + HN(7) synaptic strength due to the HN(4) interneuron on each side for each motor neuron. Note that the two sides do not differ in average proportional strength (p values: original data, and, in

*Figure 8 continued on next page*

*Figure 8 continued*

parenthesis, the mean p value after scrambling between left and right). Animal Group: Bilateral Recordings, 'Synaptics' (*Figure 2—figure supplement 1C*, dashed green circle).

DOI: https://doi.org/10.7554/eLife.31123.019

## Discussion

Across a population, the neuronal networks of individual animals arrive at unique sets of underlying parameters such as ionic conductances and synaptic strengths to achieve functional stereotypical output. This underlying variability has been well recognized and studied (*Ciarleglio et al., 2015*; *Goaillard et al., 2009*). What is less realized is that stereotypical functional output is itself variable across individuals. Our work on the leech heartbeat CPG, motor pattern and heartbeat has contributed to this realization and led to the conclusion that individual animals produce unique functional patterns at all levels – CPG, motor neurons, muscle - (*Norris et al., 2011*; *Wenning et al., 2014*; *Wright and Calabrese, 2011b*). Here we sought to determine the range of functional output to explore the potential sources of variability across individuals, and, for the first time, to compare variabilities across levels: CPG, motor neurons, hearts. We did this for both motor programs produced in leech heartbeat. We focused on phase, which is a critical output characteristic of any coordinated motor program and its underlying neuronal circuitry.

### Cycle-to-cycle phase variability in an individual

Cycle-to-cycle variability within individuals reflects imprecision in the underlying neuronal networks and muscles. This variability is thought to arise principally from the stochastic nature of all biochemical processes, particularly ion channels, synaptic release, and neuromuscular systems (*Marder and Calabrese, 1996*; *O'Leary et al., 2014*; *Roffman et al., 2012*). In leech heartbeat, the cycle-to-cycle variabilities of intersegmental phase differences ($\Delta\phi$s) were small (*Figure 3* and *Figure 3—source data 1*). Across levels and in both coordinations they were only slightly higher than that for the bilateral pair of HN(4) interneurons, which are part of the strongly, monosynaptically, and reciprocally interconnected timing network (*Figure 1*, *Figure 3A*).

Variances are reported from the gastric pattern in the isolated crab STNS within animals but this is convolved across episodes over many days (*Hamood and Marder, 2015*). In this episodic motor pattern, cycle-to-cycle variabilities were substantially larger than those in the ongoing leech heartbeat CPG pattern and motor pattern. Cycle-to-cycle variance of $\Delta\phi$ in the swim episodes of larval zebrafish seems also rather large judging from the representative example shown (*Wiggin et al., 2014*).

Many motor programs are shaped, and often stabilized, on a cycle-to-cycle basis by sensory feedback. Prominent examples are locomotion e.g., insects (*Büschges, 2005*), mastication in decapod crustaceans (*Marder et al., 2014*), and feeding in *Aplysia* (*Cullins et al., 2015a*, *Cullins et al., 2015b*). Recording from two reporter motor neurons in the *Aplysia* feeding circuit, *Cullins et al., 2015a* found that sensory feedback increases the variability within animals, i.e., decreasing stereotypy, but decreases variability across animals so that a common solution space for functional output emerges (*Cullins et al., 2015a*). Intriguingly, the variability of individual motor components is negatively correlated with their importance in behavioral performance (*Lu et al., 2015*). In leech heartbeat, the premotor HN interneurons on one side dictate a common relative timing (phase) of the premotor inputs to all ipsilateral heart motor neurons (*Figure 1*; *Maranto and Calabrese, 1984a*, *1984b*; *Norris et al., 2007a*, *2007b*). Any local sensory input to the CPG representing a segmental perturbation, if present, would thus affect all motor neurons. Our recordings of the CPG and motor neurons are in the absence of any potential feedback.

The period of leech heartbeat is modulated by a variety of inputs. Period decreases with higher metabolic demand (e.g., locomotion) and higher temperatures, and increases in higher ambient oxygen (*Arbas and Calabrese, 1984*, *1990*; *Davis, 1986*). At the same time, as in other rhythmic behaviors (e.g., lobster and crab STNS, *Bucher et al. (2005)*; *Hamood et al., 2015*; crawling in larval *Drosophila*, *Pulver et al. (2015)*; zebrafish swimming, *Masino and Fetcho, 2005*) intersegmental phase differences do not correlate with cycle period. We emphasize that this 'phase constancy' as embodied in a lack of correlation between phase and period across animals should not be construed

as suggesting that phase does not vary within individuals or indeed the lack of correlation of period and phase. In crabs, the phase relations across many gastric episodes across many days in the isolated STNS are significantly correlated with gastric frequency as seen from two individual examples (*Hamood and Marder, 2015*). In larval zebrafish, swimming occurs in episodes with declining cycle period (*Masino and Fetcho, 2005*) making this preparation ideal to determine whether intersegmental phase delay scales with the cycle period in an individual episode.

## Sources of phase variability in a motor program in a population

The population variability in the leech, at all levels, was about 2–3 times higher than the cycle-to-cycle variability (*Figure 3* and *Figure 4*). Similarly, in the pyloric pattern of the isolated STNS, variances were much lower within than across animals (*Hamood et al., 2015*). In the episodic gastric pattern, however, variances within animals, but again convolved across episodes across many days, were comparable to those across animals (*Hamood et al., 2015*). Variability increases when modulatory input is compromised (decentralization), in both the pyloric and the gastric pattern in the crab STNS (*Hamood and Marder, 2015*). In contrast, variability in interlimb phase decreases sharply in galloping mice after ablating all V0 commissural interneurons resulting in a bounding gait (*Bellardita and Kiehn, 2015*).

Two potential sources for the large variability in phase across animals at each level of the leech heartbeat system are variability in the episodic repeats of the same motor program within an individual, and variability between bilaterally homologous elements, also within an individual. Episodic variability is seen as the two alternating patterns are repeated periodically by the same neurons and muscles (*Figure 2*). Repetition variances were significantly lower than expected from the population variance across both coordinations and levels, and therefore seem not to contribute to the population variability (*Figure 6* and *Figure 6—source data 1*). Because in much of the literature episodes within an animal and across animals are convolved it is not easy to determine whether repetition variability (i.e., inter-episode variability) contributes to population variability. We suspect it does. For example, interlimb phase variance in galloping mice was assessed convolving 51 episodes across three animals so the large variance shown probably reflects large repetition variance (*Bellardita and Kiehn, 2015*). Similarly, in an elegant study on the motor and constriction patterns in fly larvae, covering forward and backward crawling, episodes and animals are convolved in the analysis presented, and the considerable inter-episode variability was not quantified (*Pulver et al., 2015*) though it certainly exists (personal communication, Stefan Pulver). In zebrafish literature again episodes and animals are convolved making it difficult to parse repetition variance and population variance (*Masino and Fetcho, 2005*; *Wiggin et al., 2014*). The heroic enterprise to record from several motor neurons in the STNS in vivo over several days (*Yarger and Stein, 2015*) does not tease apart the phase variability across episodes and across animals for the episodic gastric motor pattern.

Variability between homologous cells within an individual can also contribute to variability seen in a population. For example, in the crab heartbeat system, the five (presumably) homologous motor neurons – which are part of the CPG in this system – can express different levels of a common set of membrane conductances to achieve synchrony (*Lane et al., 2016*; *Ransdell et al., 2013*). This system seems more tightly regulated in phase than the leech heartbeat neuronal network, reflecting the synchronized rather than segment-specific nature of its motor output.

We took advantage of the bilateral layout in the leech heartbeat system to test whether variability in homologous elements can contribute to population variability. In the leech, neurons arise from bilaterally paired columns of blast cells derived from bilateral pairs of stem cells (teloblasts) each of which arises from a symmetric division. Such symmetry in the origin of bilateral blast cells gives rises to the concept of bilaterally homologous structures and cells. Blast cells differentiate into neurons, epithelia and muscles in each segment (*Stent and Weisblat, 1985*; *Weisblat and Shankland, 1985*). Yet, there is evidence for stochastic events during leech development, for example the bilateral OP neuroblasts undergo symmetric divisions but for each the fate of the daughter cells is determined by position, and also in the formation of unpaired neurons, where either the left or the right neuron dies (*Blair et al., 1990*; *Stent and Weisblat, 1985*; *Weisblat and Shankland, 1985*).

At all levels, bilaterally homologous elements (neurons and muscles) participate in heartbeat with a single CPG orchestrating two different coordinations of motor neurons and heart muscles. Bilateral variances differed depending on level (CPG, motor neurons, motor plant) and coordination (peristaltic, synchronous). In synchronous coordination, on all levels, the variance of differences in

intersegmental $\Delta\phi$ between sides ($\Delta\Delta\phi$) was significantly less than expected from the population (*Figure 7—source data 1*), and therefore unlikely to contribute to the population variability. In peristaltic coordination, however, the variance between sides ($\Delta\Delta\phi$) in the CPG pattern and in the beat pattern did not differ from the population but differed significantly from the population in the motor pattern (*Figure 7* and *Figure 7—source data 1*). Therefore, such variability between homologous elements may contribute to the population variability at least for the CPG and the beat patterns when peristaltic. This bilateral variability might arise from stochastic processes during development as envisioned e.g., in the models of O'Leary and Marder (*O'Leary et al., 2013*; *O'Leary et al., 2014*). Nevertheless, an important source of population variability is likely genetic and life history differences inherent in our population.

The sources of this high bilateral variability in the CPG and the hearts are likely to be different. For example, the bilateral homologs of the premotor interneurons of the CPG vary in the synaptic strengths of their connectivity pattern (*Roffman et al., 2012*) and likely in their intrinsic conductances. Each heart segment receives excitatory input from its ipsilateral motor neuron, these synapses may also vary in strength, and heart muscles may vary in their intrinsic conductances. Moreover, the exact timing of that heart segment's constriction appears to depend on load (*Wenning et al., 2014*), which is unlikely to be identical on both sides in a soft-bodied animal like the leech (*Wenning and Meyer, 2007*).

## Phase variability across levels of a motor system

Despite the feedforward nature of the leech heartbeat system – CPG, motor neurons, heart muscle – we found that the high phase variance in the CPG in peristaltic coordination did not translate into an equally high variance in the motor pattern (*Figure 4* and *Figure 5*). This puzzling result begs the question how the variance of the motor pattern is reduced. The peristaltic phase difference that the HE(8) to HE(12) motor neurons achieve is significantly smaller than that of the premotor interneurons of the CPG (*Figure 4* and *Figure 5*). The phase progression of the premotor bursting pattern determines the maximal phase range, but the segment-specific synaptic strength pattern, intrinsic properties, and coupling determines the phase realized between two motor neurons (*Wright and Calabrese, 2011b*). The premotor phase differences and the synaptic strength patterns interact so that the phase difference of the motor neurons progresses smoothly across the segments (*Wright and Calabrese, 2011b*), and this smoothing and segment-to-segment attenuation of the CPG phase difference may limit the variance of intersegmental $\Delta\phi$s, especially for nearby segments. In support of this conclusion, we observed that when a larger or smaller number of segments intervene between motor neurons are assessed, then the variance in the motor pattern reflects more or less the corresponding variance of the CPG (*Figure 4—figure supplement 1*). Thus, in segmentally distributed motor patterns (such as the swim networks of lampreys, fish, leeches, crayfish swimmerets and locomotor patterns in insects) it is important to define or, better even, to compare different sets of segments over which variability is assessed (*Büschges, 2005*; *Grillner and El Manira, 2015*; *Ingebretson and Masino, 2013*; *Kristan et al., 2005*; *Mullins et al., 2011*; *Pulver et al., 2015*; *Smarandache-Wellmann et al., 2014*; *Wiggin et al., 2012*).

## Conclusions

We interrogated a feed-forward motor control system to determine the output variability of phase among individuals at each level - CPG, motor neurons, muscles - and found that it varied at each level, which nevertheless did not obscure recognition of distinct coordinations. We attempted to identify some of the sources of this variability in output activity. It is unlikely due to variability in performing the same function multiple times since the repetition variances are low everywhere in the system. Some of this population variability may be due to differences between homologous cells in an individual. We observed that when the same motor act is performed by bilaterally homologous CPG neurons and heart muscles, variability in peristaltic phase on the two sides is as large as in the population itself. In other cases (peristaltic motor pattern and synchronous patterns at all levels), it appears likely that output variability is mainly associated with genetic and life history difference among individuals in the population. Across levels, phase variability was coordination-specific: similar at all levels in the synchronous but significantly lower for the motor pattern than for the CPG pattern in peristaltic coordination. Mechanisms involved in the transform from CPG to the motor neurons

may limit the range of output variability in the motor pattern. We provide a roadmap for others that may wish to analyze variability in motor system and argue that existing data sets on the locomotor and other motor patterns of invertebrates and vertebrates can be teased apart to determine the sources of output variability.

## Materials and methods

### Animals and solutions

Adult leeches (*Hirudo sp.*) were obtained from commercial suppliers (Leeches USA, Westbury, NY, or Niagara Medical Leeches (www.leeches.biz/contact) and kept in artificial pond water at 16°C. Prior to all procedures, leeches were cold-anesthetized in crushed ice for about 10 min. Dissections were done in ice-cold leech saline (composition in mM: 115 NaCl, 4 KCl, 1.8 CaCl$_2$, 10 glucose, and 10 HEPES buffer, adjusted to pH 7.4 with NaOH). Animals were superfused with leech saline during the electrophysiological experiments. For video-imaging, intact, adult leeches were flattened and covered with artificial pond water (details in *Wenning et al., 2014*). Experiments were done at room temperature (21–22°C).

### Nomenclature

In what follows ganglion and segment numbers refer to midbody segments. Body side is indicated by R and L, i.e., HE(R,8) is the heart motor neuron in segment eight on the right side, and heart (L,8) is the heart in segment eight on the left side. Referring to both sides is indicated by L/R,i.e., HN(L/R,3) refers to the bilateral pair of heart interneurons in segment 3.

We define a switch cycle as the time (or the number of neuronal bursts and heart constrictions, respectively) a given side needs to complete both coordination states, from peristaltic to synchronous to peristaltic or vice versa, one after the other. Thus, five switches are needed to record two consecutive complete switch cycles.

### Project database

The Project Database contains recordings of 153 preparations: HN interneurons in 129 animals and HE motor neurons in 83 animals (*Figure 2—figure supplement 1A*). The Project Database contains a common set (N = 59), in which both HN interneurons and HE motor neurons were recorded simultaneously (*Figure 2—figure supplement 1B*; 4-point recordings). In a subset of the HN recordings, bilateral HN interneurons were recorded (N = 17; 4-point recordings), and in a subset of the HE recordings bilateral HE motor neurons were recorded (N = 24; 4-point-recordings) (*Figure 2—figure supplement 1A*). In the common set of 59 HN/HE recordings, a subset of 9 were bilateral simultaneous recordings (*Figure 2—figure supplement 1B,C*; 8-point recordings). In this study, we report new bilateral recording of HN interneurons from a total of 26 animals and of HE motor neurons of 33 animals (*Figure 2—figure supplement 1C*). These include data on bilateral simultaneous recordings from the HE(8) and HE(14) motor neurons (N = 15) and from the HE(8) and HE(10) motor neurons (N = 9), which were recorded simultaneously with the HN(12) neurons (6-point recordings). We present new data on bilateral measurements of synaptic currents from a subset of the 26 animals in which bilateral HN recordings were made (N = 16; 'Synaptics', *Figure 2—figure supplement 1C*; 5-point recordings). We present the bilateral beat pattern for the first time from previously imaged intact adult leeches (*Wenning et al., 2014*); N = 12; *Figure 1—figure supplement 1* and *Figure 1—video 1*; 'Intact Animal Database', *Figure 2—figure supplement 1D*).

### Electrophysiological recordings and data acquisition

Electrodes were pulled on a Flaming/Brown micropipette puller (P-97, Sutter Instruments; http://www.sutter.com) from borosilicate glass (1 mm OD, 0.75 mm ID; A-M Systems; http://www.a-msystems.com).

For extracellular recordings, suction electrodes were filled with leech saline and placed in a suction electrode holder (E series, Warner Instruments; http://www.warneronline.com). To ensure a tight fit between cell and electrode, electrode tips were drawn to approximately the diameter of the cell body of the HE motor neuron (30 µm) or the HN interneuron (15 to 20 µm), respectively. The electrode tip was brought in contact with the cell body and light suction was applied until the cell body

was inside the electrode. Extracellular signals were monitored with a differential AC amplifier (model 1700, A-M Systems) at a gain of 1000 with the low- and high-frequency cutoffs set at 100 and 1,000 Hz, respectively. Noise was reduced with a 60 Hz notch filter. A second amplifier (model 410, Brownlee Precision; http://www.brownleeprecision.com) amplified the signal appropriately for digitization.

Intracellular recording techniques and voltage clamp protocols were conventional and are described in detail in (*Norris et al., 2007a*, *2011*). At the end of each voltage-clamp experiment, the electrode was withdrawn from the motor neuron. The experiment was accepted if the electrode potential was within ±5 mV of ground. Thus, holding potentials were accurate within ±5 mV.

Data were digitized (>5 kHz sampling rate), using a digitizing board (Digi-Data 1200 or 1550 Series Interface (http://www.moleculardevices.com), and acquired using pCLAMP software (http://www.moleculardevices.com) on a personal computer.

## Recording burst patterns, imaging heart constrictions, and measuring synaptic strengths

Electrophysiological recordings were done in isolated chains of ganglia. Those ganglia in which we recorded HN interneurons or HE motor neurons extracellularly were desheathed.

We recorded from the HN(L/R,4) and HN(L/R,7) interneurons in 26 animals. In two animals, only one switch cycle was recorded so we assessed the repetition variability in 24 recordings. In 9 of the 26 animals we recorded simultaneously the HE(L/R,8) and HE(L/R,12) motor neurons (*Figure 2—figure supplement 1C*).

We recorded from the HE(L/R,8) and HE(L/R,12) motor neurons in 33 animals. In one animal, only one switch cycle was recorded so we assessed the repetition variability in 32 recordings. In 9 of the 33 animals we recorded simultaneously the HN(L/R,4) and HN(L/R,7) interneurons (see above) (*Figure 2—figure supplement 1C*).

In 20 animals, we attempted to voltage-clamp 6 motor neurons, one after another: the HE(L/R,8), HE(L/R,10), and HE(L/R,12) motor neurons to determine the synaptic strength from the HN(L/R,4) and HN(L/R,7) heart interneurons recorded simultaneously. Using spike-triggered averaging (for a detailed description of the methods see *Norris et al., 2006*, *2007a*), we determined the synaptic strength in the left and right HE motor neurons (HE(8): N = 9, HE(10): N = 8, and HE(12): N = 10) in a total of 16 animals, i.e. in some animals, we successfully voltage-clamped several pairs of HE motor neurons, one after the other (*Figure 2—figure supplement 1C*).

Video-imaging of intact, adult leeches provided the optical signals to determine the bilateral beat patterns, in the motor plant (i.e., the two hearts) in midbody segments 7 to 14 (N = 12; *Figure 1—Video 1*) (*Figure 2—figure supplement 1D*). These data were compiled previously, and data acquisition and analysis were described in detail in (*Wenning et al., 2011*, *2014*). As for the motor pattern, we used segments 8 and 12. Imaging was limited to 10 min, which did not yield two complete switch cycles in all 12 animals. We assessed the repetition variability in 9 (left side) and 8 animals (right side), respectively. One animal was identified as an outlier (outside the 1.5*interquartile range on a Whisker barrel plot) because of the irregular sequence of constrictions in segments 8 and 9 on both sides in synchronous coordination.

To further examine the intersegmental phase differences and the variability across individuals, we used our current Project Database which includes unilateral recordings from the HN(4) and HN(7) interneurons and from the HE(8) and HE(12) motor neurons since bilateral recordings are not necessary for this analysis (*Figure 2—figure supplement 1A*; N = 129, CPG pattern; N = 83, motor pattern). Some of these data were published (*Norris et al., 2007a*, *2011*; *Wenning et al., 2014*) and some were presented at the Society for Neuroscience meeting in 2016 (*Norris et al., 2016*).

## Data analysis

We used specific points in time to calculate the intersegmental phase difference $\Delta\phi$ between bursts of the two pairs of HN interneurons, the two pairs of HE motor neurons, and the constrictions of the two pairs of heart segments. The detailed description of the methods and the custom-made MATLAB codes have been published (*Cymbalyuk et al., 2002*; *Masino and Calabrese, 2002c*; *Norris et al., 2007b*; *Wenning et al., 2004a*, *2004b*), so we summarize briefly here.

To characterize the bursting patterns of the HN interneurons and HE motor neurons, spikes were detected based on threshold and then grouped into bursts (interburst interval $\geq 1$ s). Stray spikes

were eliminated. The middle spike (based on count) served as the phase marker for an individual burst. We then calculated burst period ($T$) and phase ($\phi$). The burst period was defined as the interval in seconds from middle spike to middle spike of consecutive bursts (e.g. *Figure 2*, A2 for the HN premotor interneurons). Phases were referenced to an absolute phase reference ($\phi = 0$), the HN(4) interneuron on the right side. The phases of the HN(4) premotor interneuron on the left side, those of the two HN(7) premotor interneurons, and, if applicable, those of the four HE motor neurons were determined on a cycle-to-cycle basis. Phase differences (HN(R,4) - HN(R,7), HN(L 4) - HN(L,7), HE(R,8) – HE(R,12), and HE(L,8) - HE(L,12), respectively, were defined as the difference between the time of their middle spikes ($t_i$) and the time of the middle spike of the reference segment ($t_r$) in the same cycle divided by the reference cycle period ($T_r$) expressed as ($\phi_{r-i} = [(t_i - t_r)/T_r]$).

For the beat pattern of the hearts, the phase reference was the maximum rate of rise (MRR) of the digitized optical signals of an individual heartbeat cycle (*Figure 1—figure supplement 1B*, inset). The MRR corresponds to emptying (systole) and maximal heart constriction (*Wenning et al., 2014*). We determined the intersegmental $\Delta\phi$ between heart segments 8 and 12 on the left and on the right side, using the ipsilateral heart segment 8 as the phase reference.

Inhibitory postsynaptic currents (IPSCs) from the ipsilateral HN(4) and HN(7) heart interneurons were recorded in three pairs of heart motor neurons (HE(R/L,8), HE(R/L,10), and HE(R/L,12)). To normalize across different holding potentials, IPSCs were converted to, and reported as, conductances (reversal potential: −62 mV; *Angstadt and Calabrese, 1991*).

## Statistics and metrics

Angular variances were calculated by using the Cartesian average of polar phase vectors. Circular (or polar) plots were drawn, and the statistics were calculated using the Pandora Toolbox (*Günay et al., 2009*). The code, custom scripts, and the data can be found in: Sources of variability in a motor system, *Calabrese, 2018*). Using vector summation, we found the mean vector phase and length $r$ (as a measure of concentration) with 1-r as a measure of dispersion. From the vector length, we calculated the angular variance $s^2 = 2(1-r)$ (in radians squared) and from this the angular standard deviation $s$ (in radians) (*Zar, 1974*). We report these values in phase units squared or phase units by dividing by $4\pi^2$ and $2\pi$, respectively.

To assess the cycle-to-cycle variability, we used the intersegmental phase differences between two segments ($\Delta\phi$) burst-by-burst and beat-by-beat, respectively, within body side and coordination. Data on consecutive switch cycles were kept separate. This analysis yielded 4 data sets per body side for each level (peristaltic 1, synchronous 1, peristaltic 2, synchronous 2); 24 sets total.

To assess the population variability, we used the average intersegmental $\Delta\phi$ of a single switch cycle on each level, for each side and coordination. To assess whether these population variances differed between levels we resampled the data (with replacement; 10,000 times; 'bootstrapping'), calculated the 95% confidence intervals, and compared variances between two levels. This analysis yielded 4 data sets for each level, 12 sets total.

To assess the repetition variability, we first determined whether the cycle-to-cycle intersegmental $\Delta\phi$ on one side in a given coordination differed between two consecutive switch cycles (unpaired t-test). To compare the repetition variability across animals, we calculated the difference of the average intersegmental $\Delta\phi$ between two consecutive switch cycles (the $\Delta$ of $\Delta\phi$), within coordination and side. Note that repetitions of the same coordination are separated by the time it takes to execute the other coordination, about 2–4 min. For example, for the CPG we calculated

$$\Delta\Delta\phi = \Delta\phi_{\text{cycle1}}(\text{HN(L,4)} - \text{HN(L,7)}) - \Delta\phi_{\text{cycle2}}(\text{HN(L,4)} - \text{HN(L,7)})$$

We plotted the individual $\Delta\Delta\phi$ in a circular phase plot and calculated their variance. This analysis yielded 4 data sets for each level, 12 sets total. We repeated this procedure 10,000 times after randomly combining (scrambling) two switch cycles from different animals, and calculated the p values from the z score of the normal distribution of the scrambled populations. The motor pattern and the beat pattern were treated the same way.

To assess the bilateral variability, we used one switch cycle. We first determined whether the cycle-to-cycle intersegmental $\Delta\phi$ differed between the two sides (unpaired t-test). To compare the bilateral variability across animals, we calculated the difference of the average intersegmental

$\Delta\phi$ within coordination between the right and the left side (the $\Delta$of $\Delta\phi$). Note that the two sides do not execute the same coordination at the same time. For the CPG, we calculated

$$\Delta\Delta\phi = \Delta\phi_{\mathrm{right}}(\mathrm{HN(L,4)} - \mathrm{HN(L,7)}) - \Delta\phi_{\mathrm{left}}(\mathrm{HN(L,4)} - \mathrm{HN(L,7)})$$

We plotted the individual $\Delta\Delta\phi$ in a circular phase plot, and calculated their variance. This analysis yielded two data sets (one for each coordination) per level, 6 sets total. We repeated this procedure 10,000 times after scrambling the data from the right and left body sides from different animals, and calculated the p values from the z score of the normal distribution of the scrambled populations. The motor pattern and the beat pattern were treated the same way.

We assessed the variability of the synaptic input from premotor heart interneurons HN(R,4) and HN(R,7) to heart motor neurons HE(R,8), HE(R,10) and HE(R,12), and from heart interneurons HN (L,4) and HN(L,7) to heart motor neurons HE(L,8), HE(L,10) and HE(L,12). This analysis yielded two data sets per motor neuron pair, 6 sets total (*Figure 8A*). To evaluate the synaptic input further and to eliminate any differences due to the quality of the voltage-clamp recordings, we calculated the proportion of the synaptic strength due to the HN(4) premotor interneuron on each side using its ratio to the sum of the synaptic inputs of both premotor interneurons HN(4)/[HN(4) + HN(7)], separately for each side (*Figure 8B*). We used a paired t-test to assess whether the two sides were different. Next, we scrambled the data from the right and left body sides from different animals (10,000 times), and calculated their p values using a paired t-test.

We used means ±SD for descriptive statistics.

## Acknowledgements

Dr. Anca Doloc-Mihu significantly improved the code to analyze spike-triggered averages and Florencia Zamora helped with the analysis. We would like to thank Dr. Mark A Masino for his comments on an earlier version of this manuscript. Supported by NIH NINDS 1 R01 NS085006 to RLC.

## Additional information

### Competing interests
Ronald L Calabrese: Reviewing editor, *eLife*. The other authors declare that no competing interests exist.

### Funding

| Funder | Grant reference number | Author |
| --- | --- | --- |
| National Institute of Neurological Disorders and Stroke | 1 R01 NS085006 | Angela Wenning<br>Brian J Norris<br>Cengiz Günay<br>Daniel Kueh<br>Ronald L Calabrese |

The funders had no role in study design, data collection and interpretation, or the decision to submit the work for publication.

### Author contributions
Angela Wenning, Conceptualization, Data curation, Formal analysis, Validation, Investigation, Visualization, Methodology, Writing—original draft, Project administration, Writing—review and editing; Brian J Norris, Conceptualization, Validation, Investigation, Methodology; Cengiz Günay, Software, Methodology; Daniel Kueh, Validation, Methodology; Ronald L Calabrese, Conceptualization, Resources, Funding acquisition, Validation, Visualization, Project administration, Writing—review and editing

## Author ORCIDs

Angela Wenning (ID) http://orcid.org/0000-0002-9400-6280
Cengiz Günay (ID) http://orcid.org/0000-0001-7586-571X
Daniel Kueh (ID) http://orcid.org/0000-0002-2462-5203
Ronald L Calabrese (ID) http://orcid.org/0000-0001-7135-3469

## Decision letter and Author response

Decision letter https://doi.org/10.7554/eLife.31123.025
Author response https://doi.org/10.7554/eLife.31123.026

## Additional files

### Supplementary files

• Supplementary file 1. Experimental data and analysis for all figures.
DOI: https://doi.org/10.7554/eLife.31123.020

• Transparent reporting form
DOI: https://doi.org/10.7554/eLife.31123.021

### Major datasets

The following dataset was generated:

| Author(s) | Year | Dataset title | Dataset URL | Database, license, and accessibility information |
|---|---|---|---|---|
| Ronald L Calabrese | 2018 | Sources of variability in a motor system | http://dx.doi.org/10.5061/dryad.c0g0p | Available at Dryad Digital Repository under a CC0 Public Domain Dedication |

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
