## [Decision Letter]

[Editors’ note: a previous version of this study was rejected after peer review, but the authors submitted for reconsideration. The first decision letter after peer review is shown below.]

Thank you for submitting your work entitled "Sources of variability in a motor system" for consideration by *eLife*. Your article has been reviewed by three peer reviewers, one of whom is a member of our Board of Reviewing Editors and the evaluation has been overseen by a Senior Editor.

Our decision has been reached after consultation between the reviewers. Based on these discussions and the individual reviews below, we regret to inform you that your work will not be considered further for publication in *eLife*. Nonetheless, if you feel that a significant rewrite could address the bulk of the reasonable issues raised by the reviewers (including some of the parts of the consultation session included here), we would be willing to consider a new version along those lines.

This manuscript and its reviews engendered a long and spirited discussion among the reviewers, the Reviewing Editor and the Senior Editor.

The Reviewing Editor's initial summary was: "While all reviewers saw the value of this meta-analysis, they also missed the experimental approach that brings this beyond the descriptive level. The reviewers also didn't see a quick fix to add critical experiments to link the data in a meaningful manner. They also felt that the authors did not attempt to appeal to a general readership which would make this study more attractive for *eLife*."

In looking back at the extended discussions among the reviewers, it is clear that:

a) It was difficult to figure out which data came from the same animals, and therefore really supported the up and down level of organization point you were trying to make.

b) The manuscript itself is very invertebrate-centric and there may be useful and informative vertebrate work that should be cited.

c) The manuscript has to proactively argue, at least in a good paragraph in the Discussion section, why you believe that the variability is not due to experimentor-error.

d) It is not clear that the stats are properly done or presented (SEs versus SDs, etc.).

e) The argument about developmental stochasticity being responsible for left/right differences is very speculative and should be relegated to a Discussion point but not elevated to the extent it is in the present version.

f) The Abstract was hard to parse.

g) Is it really the case that the variability increases and decreases across the levels of organization? If so, is that an interesting result that needs a possible explanation? This finding is probably part of what motivated a reviewer to worry/wonder about experimental error.

We understand that it may not be possible to do a manipulation that would address the Reviewing Editor’s initial evaluation, but points a-g are clearly fixable.

Some of the comments made during discussion were:

"But there **could** be controlled measurements in their data set. I don't think they need measurements of all levels in the same animals – even if they led us on a transitive journey of dual measurements where A+B and B+C told us something about variability in A relative to C. Without these (which are definitely not in this version of the manuscript), every bias against the variability story will come into play and every can of worms will be opened as people read this study. "

"First, the premise about developmental stochasticity is nothing more than a speculation and should be introduced as that in a much more restricted fashion.

Second, many, if not all of the standard errors should be standard deviations. More critically, I am not sure exactly what was used as N's to calculate p values. there appear to be some very very small p's that may have resulted from using number of cycles not animals. So, I am not sure that all of the stats are correctly done."

"I agree that the paper is highly invertebrate-centric, but I am not sure whether a similar data exists in a vertebrate? If so, they should certainly consider it."

"Data such as these that clearly pose the issue of whether variability at one level is transmitted to another are rare anywhere, and if properly presented, I think this would be a valuable contribution…What I find intriguing is that at face value the variability does not change monotonically across levels, and they don't really address that adequately."

"I guess I just think that this paper is a very brave attempt to do something that needs to be done, and with all of the issues of the present version, there is nothing quite like it in the literature and it is addressing a very interesting problem."

"I am not as convinced that it is possible to parse what is experimentally induced variation versus biological variation in the various comparisons in the paper. I wish it had actually been addressed head on in the paper."

"I guess my problem was/is that I have/had a preconceived idea what to expect from a paper dealing with variability, and I didn't see this in the manuscript -hence my negative reaction.

In mammalian rhythm generating networks (e.g. neocortex and of course the respiratory network) you find a huge amount of cycle-to-cycle variability which is the reflection of the stochastic activation/synchronization of weakly coupled neurons. And also, we find of course a huge variability in the types of inward/outward currents in rhythm generating neurons. Thus, variability is a big deal in how these networks operate and studying this variability using computational approaches, gives you interesting insights into the network architecture. Variability is also reflected into how the network oscillation is transmitted to the motor output.

Anyhow, we and other used experimental and computational approaches to dissect the mechanisms for this variability and how it assembles these rhythm generating networks. All this is built on the wonderful work that has been done in invertebrates. I didn't see anything of this in this paper. This paper appeared to me like a big collection of variability data that are found at different levels of integration – but I didn't see the big picture question."

Reviewer #1:

This study addresses an important issue in neurobiology: i.e. the nature of cycle-by-cycle variability in rhythm generating neuronal networks. The authors are surprisingly agnostic to the literature in mammalian rhythm generating networks, and focus primarily on the leech heart beat system with some reference to other invertebrate rhythm generating systems. This omission unnecessarily limits the scope of this fundamentally, important issue.

The authors base their study on a larger data set, which allows them to compare sources of variability at various levels of integration from the cellular level to the behavioral level. However, one major weakness of this meta-analysis is the absence of experimental attempts to explore the actual mechanisms that underlie this stochastic variability, and the question how variability at one level influences another level.

The dataset presented in this manuscript seems ideal for a subsequent computational approach, which could model the detailed data to explore how variability at one level influences variability at another level of integration. This would be one kind of "experimental approach" that could be very informative and would likely provide interesting mechanistic insights and testable hypotheses. But, without a computational approach or actual experiments in this ideal model system, the study remains very descriptive.

Reviewer #2:

The authors have compiled an extensive and impressive resource of data for motor pattern output across three levels of organization for a CPG network responsible for heart contraction in the leech. In a thorough and insightful analysis, they reveal that output variability across individuals, as measured by phase relationships, is different at different levels of the feedforward circuit, with the least variability seen at the level of the motor network, as opposed to the CPG or musculature. This is a compelling insight.

I do have serious concerns that I feel need to be addressed:

1) The nature and scope of the data themselves, and the nature and scope of the interpretation and discussion, are not in balance. Even from the title, "Sources of variability[…]" this imbalance is established, as I do not feel the authors are in any way assessing the mechanistic underpinnings or sources of variability. In particular, the use of bilateral measurements – and the demonstration therein of differences – is held up as evidence of "stochasticity in developmental pathways" among genetically identical cells and networks. One cannot argue with the genetics in question, but there is no evidence whatsoever to support this hypothesis. Indeed, inherent in this perspective is a bias that these are truly bilaterally symmetrical. Is there any data to support this assertion, or is this just an assumption? One could make the case that at a superficial level the mammalian heart is bilaterally symmetrical as well, but physiologically the pulmonary and systemic circuits are clearly differentiated in their function and underlying physiology. If this is the focus the authors wish to take, there needs to be a substantial redirection of the introduction and discussion of this work to try to make this case. Else the authors may have stumbled on to compelling physiological evidence to call into question the idea that the annelid heart is indeed a bilaterally symmetrical structure (at this point, an equally parsimonious – if not more so – interpretation of the data). In short, I do not feel that anywhere in the manuscript the authors make a compelling enough case to draw any conclusions about sources of variability. Phenomenologically, the variability is clear – but the etiology is completely unexplored in this study.

2) Variability is perhaps one of the more difficult phenomena to quantify, due to its – well, variability. Because the data in this study are substantially meta-analytical, there is substantive lack of experimental control over potential sources of variability that preclude combining these data into one analysis. For example, comparing the amount of variability in measurements of muscle output for data collected multiple years ago with a cohort of animals entirely distinct of that from which the motor neuron output was collected is potentially confounding. The authors utilize animals supplied from multiple vendors outside of their own control (are they wild caught or reared?). How can one ensure rearing conditions, environmental variables, etc. are consistent between vendors – or across time – to appropriately cross-compare these different data sets into one analysis? Is there any potential species confusion across the time span of these studies? Indeed, the species used is listed only as "Hirudo sp." and recently Hirudo has undergone a species reclassification. if the authors may be inadvertently, and more importantly asymmetrically, combining data from multiple species this is highly problematic. The most compelling data are those in which multiple levels of analysis are achieved from the same individuals – statistics for these are reported as a sub-population in the text, but to me the most informative analysis would be a correlation of variability in, for example, motor neuron variability and muscle output variability across individuals in which data were obtained for both. The authors do apparently have 9 instances of paired CPG and motor neuron recordings, as well as simultaneous motor and beat pattern from an earlier study (Wenning et al., 2014). By lumping these data together as populations, and not controlling the abovementioned sources of variability among individuals, the authors lose power for their interpretation.

3) There is an unexplored aspect to these data that I would have appreciated hearing the authors' perspective on. That is, what are the functional implications – costs, tradeoffs, etc. – for differing variability across these levels? In other words, why should the system require highest constancy of phase at the level of motor neurons? And just as interesting, how is this achieved – how does a system take feedforward input from a CPG and decrease the variability in that input signal? I am keenly interested to hear the authors' insight on this mechanistic question, and its functional consequences – and I feel this is the more compelling aspect of the data presented.

In sum, I really enjoyed the thorough analysis of these data and the collection and curation therein that allows for interesting comparisons across levels of organization in a feedforward system. I am less convinced that pitching this work in the context of "sources of variability" is congruent with the data at hand. The challenges associated with a meta-analysis in this context are significant.

Reviewer #3:

This paper explores an important question concerning how variable a motor output is within and across animals at the level of the CPG, the motor output, and the behavior. There is a widespread notion that there are many solutions to producing essentially the same motor output. One might be led to think that there is not great variability of output across animals or within animals, but a critical analysis of this question depends on strong data sets that compare patterns across levels from cells to behavior and across individuals in the same motor behavior. This paper does that for the heart beat rhythm in leeches. The major take home message is that, while there is relatively low variation in phase relationships on a cycle-to-cycle basis in a pattern produced by the heart circuit on one side of an individual animal, there is substantial variation at levels from CPG to behavior across animals. So, the output can be variable, though presumably still in an appropriately functional range. Importantly, even on opposite sides in an individual animal, the separate rhythm generators for heartbeat on the two sides can show as much difference as that seen between animals. This suggests that the assembly of the networks leads to differences in the motor patterns that may underlie observed variations. In short, rhythms across animals (and even within animals on the two sides) are not as consistent as we might think from cellular to behavioral levels, but one presumes that they are in a range that is good enough. This is important data for coming to grips with what variability at cellular and `molecular levels means for variability in motor behavior. Except for a few places the paper offers a clear verbal account and an especially clear graphical presentation of the data.

Some concerns:

1) The Abstract is poor. I read it initially and could not really figure out what it was talking about. The paper itself is way better and the figures are excellent, but the abstract needs to be done in a way that makes the conclusions easily accessible to a person reading it. To be more concrete, in the parts starting on line 16, it is not obvious what repetition variability means, and the sentence after that one is also hard to parse without having read the paper.

2) I thought that the last two sentences of the subsection “Conclusions” seemed to be a side issue not appropriate for the big conclusions.

3) My biggest issue with the work however is the potential for variability that is the result of experimental manipulation necessary to collect the data. The preparations at some levels involve dissections and even some compression of the animal for imaging the beats. There is no account of how that might affect variation across preparations. Could the cross individual variation have to do with varying levels of damage to the preparations rather than a difference in the animals themselves. Even within an animal, the two sides might different because of some differences in the level of experimental damage within the two different networks that can still produce a stable output, but with a shifted phase. I am not sure exactly how to rule this out, but it seems to be something that deserves consideration in the paper. Is there is good reason to think it is not the source of the variation? And if it is, the paper may speak less to the construction of networks and perhaps more to their destruction.

[Editors’ note: what now follows is the decision letter after the authors submitted for further consideration.]

Thank you for resubmitting your work entitled "Output variability across animals and levels in a motor system" for further consideration at *eLife*. Your revised article has been favorably evaluated by Eve Marder (Senior editor), a Reviewing editor, and three reviewers, one of whom is a member of our Board of Reviewing Editors.

The manuscript has been substantially improved, but there are some remaining issues that need to be addressed editorially before acceptance, as outlined below:

The reviewers agree that is a fascinating paper demonstrating that the leech heartbeat system exhibits remarkable intra- and interindividual variability, and that variability is found at every level from the cellular to the behavioral level. As the authors state, variability across all levels is a hallmark of neuronal networks and behavior, and it is remarkable that even a small neuronal network with identified neurons and a known connectivity shows significant variability. The finding that each animal arrives at a unique solution of producing a heartbeat motor pattern, but that sometimes the intraindividual variability is larger than the variability across individuals is of general interest. The leech heartbeat system is an ideal system to study this topic, since it is not "stabilized" by sensory inputs. The analyses collected over many years provide a rich groundwork for not only the origins of, but also the implications for, variability in motor networks in general.

We still believe that the manuscript could be further improved by discussing and introducing the data from a broader perspective. The following suggestions may help:

1) We appreciate the developmental angle as one possible explanation for the variability. The readership would benefit even more from this possible explanation, if the authors would better emphasize that homologous neurons arise from bilateral homologous neuroblasts. It's in there, but it is more alluded to than directly stated. It is a fascinating fact that bilateral neurons are not homologous but actually analogous. A finding itself that would be quite interesting for a general reader. If there is firm developmental evidence that these are, in fact, truly bilateral homologues then a more deliberate statement and citation of the evidence would go a long way to shoring up the "genetic/life history" hypothesis of the origins of the variability.

2) The very last sentence of the paper could more overtly place the charge on those model systems that are able to keenly control genetics and life history to take up the mantle of this idea next (I'm looking at you, *Drosophila* larval circuit people). If the Discussion/Conclusion section could draw from *Drosophila* or vertebrate literature in some way to show the groundwork for this charge, it would make for a compelling closing statement. Perhaps there is something in the recent beautiful work of the El-Manira group in zebrafish or *Drosophila* work by Landgraf or Pulver that isn't overtly discussed as variability, but to a trained eye has all the hallmarks. I'm not sure if there is fruit to be picked there, but if there were it would make a potent closing statement about where this foundational work could be extended to model systems with more "tools" at their disposal.

There is also ample literature out there on gait variability and the clinical implications in the context of neurological disorders or ageing. There is also lots of mammalian literature that tries to decrease intraindividual variability with sensory stimulation. The authors have a great opportunity to emphasise the unique advantage of the leech heartbeat system in tackling this very relevant topic. For scientists working in mammals it is very perplexing that "even" in these small networks every animal finds its own solution to set up the rhythmic pattern. This is a fascinating take home message.

3) While we would like to ask you for a bigger picture discussion, the paper would benefit from reducing the Discussion section, by trying to make it more concise.

---

## [Author Response]

[Editors’ note: the author responses to the first round of peer review follow.]

Reviewer #1:This study addresses an important issue in neurobiology: i.e. the nature of cycle-by-cycle variability in rhythm generating neuronal networks. The authors are surprisingly agnostic to the literature in mammalian rhythm generating networks, and focus primarily on the leech heart beat system with some reference to other invertebrate rhythm generating systems. This omission unnecessarily limits the scope of this fundamentally, important issue.

We were not interested per se in cycle-to-cycle variability; we explore it here mainly to underpin our observations about differences among preparations, across levels, between episodic repetitions, and among bilaterally homologous neurons. We have not found corresponding studies in vertebrate rhythmic motor systems, except perhaps birdsong, which we made every effort to incorporate into Discussion section.

The authors base their study on a larger data set, which allows them to compare sources of variability at various levels of integration from the cellular level to the behavioral level. However, one major weakness of this meta-analysis is the absence of experimental attempts to explore the actual mechanisms that underlie this stochastic variability, and the question how variability at one level influences another level.

We rely on extensive experimental analyses we have made over the years on the mechanisms of patterning within the heartbeat system and use this knowledge to inform our analyses. We have made an attempt to explore how variability can differ between levels with new analyses (Figure 5, Figure 6 and Figure 5—figure supplement 1).

The dataset presented in this manuscript seems ideal for a subsequent computational approach, which could model the detailed data to explore how variability at one level influences variability at another level of integration. This would be one kind of "experimental approach" that could be very informative and would likely provide interesting mechanistic insights and testable hypotheses. But, without a computational approach or actual experiments in this ideal model system, the study remains very descriptive.

Please see:

Weaver et al., 2010, Wright and Calabrese, 2011 and Wright and Calabrese, 2011.

Reviewer #2:The authors have compiled an extensive and impressive resource of data for motor pattern output across three levels of organization for a CPG network responsible for heart contraction in the leech. In a thorough and insightful analysis, they reveal that output variability across individuals, as measured by phase relationships, is different at different levels of the feedforward circuit, with the least variability seen at the level of the motor network, as opposed to the CPG or musculature. This is a compelling insight.I do have serious concerns that I feel need to be addressed:1) The nature and scope of the data themselves, and the nature and scope of the interpretation and discussion, are not in balance. Even from the title, "Sources of variability[…]" this imbalance is established, as I do not feel the authors are in any way assessing the mechanistic underpinnings or sources of variability. In particular, the use of bilateral measurements – and the demonstration therein of differences – is held up as evidence of "stochasticity in developmental pathways" among genetically identical cells and networks. One cannot argue with the genetics in question, but there is no evidence whatsoever to support this hypothesis. Indeed, inherent in this perspective is a bias that these are truly bilaterally symmetrical. Is there any data to support this assertion, or is this just an assumption? One could make the case that at a superficial level the mammalian heart is bilaterally symmetrical as well, but physiologically the pulmonary and systemic circuits are clearly differentiated in their function and underlying physiology. If this is the focus the authors wish to take, there needs to be a substantial redirection of the introduction and discussion of this work to try to make this case. Else the authors may have stumbled on to compelling physiological evidence to call into question the idea that the annelid heart is indeed a bilaterally symmetrical structure (at this point, an equally parsimonious – if not more so – interpretation of the data). In short, I do not feel that anywhere in the manuscript the authors make a compelling enough case to draw any conclusions about sources of variability. Phenomenologically, the variability is clear – but the etiology is completely unexplored in this study.

We have scaled back our claims about pinpointing the sources of variability in the population (note change in Title and Abstract), and we speculate about the origin of bilateral variability in the Discussion section “This bilateral variability might arise from stochastic processes during development as envisioned e.g., in the models of O’Leary and Marder, (2013, 2014).” We think that the reviewer has been confused by our complicated ‘simple’ system. There are bilaterally paired hearts that are identical to the eye. They both perform peristalsis alternating with synchrony in a reciprocal manner; left heart peristaltic/right heart synchronous and vice versa. They do vary in the corresponding (peristaltic/synchronous) patterns they produce across sides (as they do among individuals) but there is no handedness that we can see. The nervous elements that control the hearts consist of bilaterally paired motor neurons and interneurons. These produce reciprocally switching peristaltic and synchronous patterns corresponding to the heartbeat. There is variability in the corresponding (peristaltic/synchronous) patterns they produce across sides (as they do among individuals) but again there is no handedness that we can see. The development of the leech nervous system has been studied in great detail and these studies suggest that bilateral homologous neurons arise from bilateral homologous neuroblasts. Symmetry is rampant in the leech nervous system.

2) Variability is perhaps one of the more difficult phenomena to quantify, due to its – well, variability. Because the data in this study are substantially meta-analytical, there is substantive lack of experimental control over potential sources of variability that preclude combining these data into one analysis. For example, comparing the amount of variability in measurements of muscle output for data collected multiple years ago with a cohort of animals entirely distinct of that from which the motor neuron output was collected is potentially confounding. The authors utilize animals supplied from multiple vendors outside of their own control (are they wild caught or reared?). How can one ensure rearing conditions, environmental variables, etc. are consistent between vendors – or across time – to appropriately cross-compare these different data sets into one analysis? Is there any potential species confusion across the time span of these studies? Indeed, the species used is listed only as "Hirudo sp." and recently Hirudo has undergone a species reclassification. if the authors may be inadvertently, and more importantly asymmetrically, combining data from multiple species this is highly problematic. The most compelling data are those in which multiple levels of analysis are achieved from the same individuals – statistics for these are reported as a sub-population in the text, but to me the most informative analysis would be a correlation of variability in, for example, motor neuron variability and muscle output variability across individuals in which data were obtained for both. The authors do apparently have 9 instances of paired CPG and motor neuron recordings, as well as simultaneous motor and beat pattern from an earlier study (Wenning et al., 2014). By lumping these data together as populations, and not controlling the abovementioned sources of variability among individuals, the authors lose power for their interpretation.

We have thoroughly addressed this issue and have added a figure (Figure 2) that fully explains all the groups of animals that we use and we constantly refer to this figure in the text to make sure that the reader know which animals we are referring to. All Ns in the manuscript refer to the number of animals (preparations).

3) There is an unexplored aspect to these data that I would have appreciated hearing the authors' perspective on. That is, what are the functional implications – costs, tradeoffs, etc. – for differing variability across these levels? In other words, why should the system require highest constancy of phase at the level of motor neurons? And just as interesting, how is this achieved – how does a system take feedforward input from a CPG and decrease the variability in that input signal? I am keenly interested to hear the authors' insight on this mechanistic question, and its functional consequences – and I feel this is the more compelling aspect of the data presented.

We now attempt to explain how variability translates from level to level and make new analyses to elucidate how known mechanisms of circuit operation may explain this translation of variability (Figure 5, Figure 5—figure supplement 1, Figure 6).

Reviewer #3:This paper explores an important question concerning how variable a motor output is within and across animals at the level of the CPG, the motor output, and the behavior. There is a widespread notion that there are many solutions to producing essentially the same motor output. One might be led to think that there is not great variability of output across animals or within animals, but a critical analysis of this question depends on strong data sets that compare patterns across levels from cells to behavior and across individuals in the same motor behavior. This paper does that for the heart beat rhythm in leeches. The major take home message is that, while there is relatively low variation in phase relationships on a cycle-to-cycle basis in a pattern produced by the heart circuit on one side of an individual animal, there is substantial variation at levels from CPG to behavior across animals. So, the output can be variable, though presumably still in an appropriately functional range. Importantly, even on opposite sides in an individual animal, the separate rhythm generators for heartbeat on the two sides can show as much difference as that seen between animals. This suggests that the assembly of the networks leads to differences in the motor patterns that may underlie observed variations. In short, rhythms across animals (and even within animals on the two sides) are not as consistent as we might think from cellular to behavioral levels, but one presumes that they are in a range that is good enough. This is important data for coming to grips with what variability at cellular and `molecular levels means for variability in motor behavior. Except for a few places the paper offers a clear verbal account and an especially clear graphical presentation of the data.Some concerns:1) The Abstract is poor. I read it initially and could not really figure out what it was talking about. The paper itself is way better and the figures are excellent, but the abstract needs to be done in a way that makes the conclusions easily accessible to a person reading it. To be more concrete, in the parts starting on line 16, it is not obvious what repetition variability means, and the sentence after that one is also hard to parse without having read the paper.

We have completely rewritten the Abstract and hope it is clearer.

2) I thought that the last two sentences of the subsection “Conclusions” seemed to be a side issue not appropriate for the big conclusions.

Fixed.

3) My biggest issue with the work however is the potential for variability that is the result of experimental manipulation necessary to collect the data. The preparations at some levels involve dissections and even some compression of the animal for imaging the beats. There is no account of how that might affect variation across preparations. Could the cross individual variation have to do with varying levels of damage to the preparations rather than a difference in the animals themselves. Even within an animal, the two sides might different because of some differences in the level of experimental damage within the two different networks that can still produce a stable output, but with a shifted phase. I am not sure exactly how to rule this out, but it seems to be something that deserves consideration in the paper. Is there is good reason to think it is not the source of the variation? And if it is, the paper may speak less to the construction of networks and perhaps more to their destruction.

We forthrightly address this issue in our letter of appeal. There is only ONE CPG network consisting of bilaterally paired neurons; if one side is messed up, the other is too.

[Editors' note: the author responses to the re-review follow.]

Thank you for resubmitting your work entitled "Output variability across animals and levels in a motor system" for further consideration at eLife. Your revised article has been favorably evaluated by Eve Marder (Senior editor), a Reviewing editor, and three reviewers, one of whom is a member of our Board of Reviewing Editors.The manuscript has been substantially improved, but there are some remaining issues that need to be addressed editorially before acceptance, as outlined below:The reviewers agree that is a fascinating paper demonstrating that the leech heartbeat system exhibits remarkable intra- and interindividual variability, and that variability is found at every level from the cellular to the behavioral level. As the authors state, variability across all levels is a hallmark of neuronal networks and behavior, and it is remarkable that even a small neuronal network with identified neurons and a known connectivity shows significant variability. The finding that each animal arrives at a unique solution of producing a heartbeat motor pattern, but that sometimes the intraindividual variability is larger than the variability across individuals is of general interest. The leech heartbeat system is an ideal system to study this topic, since it is not "stabilized" by sensory inputs. The analyses collected over many years provide a rich groundwork for not only the origins of, but also the implications for, variability in motor networks in general.We still believe that the manuscript could be further improved by discussing and introducing the data from a broader perspective. The following suggestions may help:1) We appreciate the developmental angle as one possible explanation for the variability. The readership would benefit even more from this possible explanation, if the authors would better emphasize that homologous neurons arise from bilateral homologous neuroblasts. It's in there, but it is more alluded to than directly stated. It is a fascinating fact that bilateral neurons are not homologous but actually analogous. A finding itself that would be quite interesting for a general reader. If there is firm developmental evidence that these are, in fact, truly bilateral homologues then a more deliberate statement and citation of the evidence would go a long way to shoring up the "genetic/life history" hypothesis of the origins of the variability.

We clarified the operational definition of bilaterally homologous neurons in leeches and clarified our arguments about the stochastic nature of bilateral differences.

Subsection “Sources of phase variability in a motor program in a population”:

We took advantage of the bilateral layout in the leech heartbeat system to test whether variability in homologous elements can contribute to population variability. In the leech, neurons arise from bilaterally paired columns of blast cells derived from bilateral pairs of stem cells (teloblasts) each of which arises from a symmetric division. Such symmetry in the origin of bilateral blast cells gives rises to the concept of bilaterally homologous structures and cells. Blast cells differentiate into neurons, epithelia and muscles in each segment (Stent and Weisblat, 1985; Weisblat and Shankland, 1985). Yet, there is evidence for stochastic events during leech development, for example the bilateral OP neuroblasts undergo symmetric divisions but for each the fate of the daughter cells is determined by position, and also in the formation of unpaired neurons, where either the left or the right neuron dies (Blair et al., 1990; Stent and Weisblat, 1985; Weisblat and Shankland, 1985).

2) The very last sentence of the paper could more overtly place the charge on those model systems that are able to keenly control genetics and life history to take up the mantle of this idea next (I'm looking at you, Drosophila larval circuit people). If the Discussion/Conclusion section could draw from Drosophila or vertebrate literature in some way to show the groundwork for this charge, it would make for a compelling closing statement. Perhaps there is something in the recent beautiful work of the El-Manira group in zebrafish or Drosophila work by Landgraf or Pulver that isn't overtly discussed as variability, but to a trained eye has all the hallmarks. I'm not sure if there is fruit to be picked there, but if there were it would make a potent closing statement about where this foundational work could be extended to model systems with more "tools" at their disposal.There is also ample literature out there on gait variability and the clinical implications in the context of neurological disorders or ageing. There is also lots of mammalian literature that tries to decrease intraindividual variability with sensory stimulation. The authors have a great opportunity to emphasise the unique advantage of the leech heartbeat system in tackling this very relevant topic. For scientists working in mammals it is very perplexing that "even" in these small networks every animal finds its own solution to set up the rhythmic pattern. This is a fascinating take home message.

We have now explored the literature for *Drosophila*, zebrafish, mice, and others and indicated where the potential exists for a meta-analysis of the type we have done (Discussion section). Indeed, the data are there (at least in the STNS, zebrafish, fly larval crawling) – but the variability across episodes and across animals have not been teased apart.

We also extended the discussion on the mechanisms by which the variability in the motor pattern is reduced.

Subsection “Phase variability across levels of a motor system”:

This puzzling result [less variance in the motor pattern] begs the question how the variance of the motor pattern is reduced. The peristaltic phase difference that the HE(8) to HE(12) motor neurons achieve is significantly smaller than that of the premotor interneurons of the CPG (Figure 4 and Figure 5). The phase progression of the premotor bursting pattern determines the maximal phase range, but the segment-specific synaptic strength pattern, intrinsic properties, and coupling determines the phase realized between two motor neurons (Wright and Calabrese, 2011). The premotor phase differences and the synaptic strength patterns interact so that the phase difference of the motor neurons progresses smoothly across the segments, and this smoothing and segment-to-segment attenuation of the CPG phase difference may limit the variance of intersegmental Dfs, especially for nearby segments. In support of this conclusion, we observed that when a larger or smaller number of segments intervene between motor neurons are assessed, then the variance in the motor pattern reflects more or less the corresponding variance of the CPG (Figure 4—figure supplement 1). Thus, in segmentally distributed motor patterns (such as the swim networks of lampreys, fish, leeches, crayfish swimmerets and locomotor patterns in insects) it is important to define or, better even, to compare different sets of segments over which variability is assessed (Buschges, 2005; Grillner and El Manira, 2015; Ingebretson and Masino, 2013; Kristan et al., 2005; Mullins et al., 2011; Pulver et al., 2015; Smarandache-Wellmann et al., 2014; Wiggin et al., 2012).

Discussion section very last sentence:

We provide a roadmap for others that may wish to analyze variability in motor system and argue that existing data sets on the locomotor and other motor patterns of invertebrates and vertebrates can be teased apart to determine the sources of output variability.

3) While we would like to ask you for a bigger picture discussion, the paper would benefit from reducing the Discussion section, by trying to make it more concise.

We shortened the original discussion considerably but added back considerations about other systems. Cut about 10%.